**EMBO** *reports*

# An integrated anatomical, functional and evolutionary view of the *Drosophila* olfactory system

Richard Benton [1,5✉], Jérôme Mermet [1,5], Andre Jang [2], Keita Endo [3], Steeve Cruchet[1] & Karen Menuz [2,4✉]

## Abstract

**The *Drosophila melanogaster* olfactory system is one of the most intensively studied parts of the nervous system in any animal. Composed of ~50 independent olfactory neuron classes, with several associated hygrosensory and thermosensory pathways, it has been subject to diverse types of experimental analyses. However, synthesizing the available information is limited by the incomplete data and inconsistent nomenclature found in the literature. In this work, we first "complete" the peripheral sensory map through the identification of a previously uncharacterized antennal sensory neuron population expressing Or46aB, and the definition of an exceptional "hybrid" olfactory neuron class comprising functional Or and Ir receptors. Second, we survey developmental, anatomical, connectomic, functional, and evolutionary studies to generate an integrated dataset and associated visualizations of these sensory neuron pathways, creating an unprecedented resource. Third, we illustrate the utility of the dataset to reveal relationships between different organizational properties of this sensory system, and the new questions these stimulate. Such examples emphasize the power of this resource to promote further understanding of the construction, function, and evolution of these neural circuits.**

**Keywords** *Drosophila melanogaster*; Olfaction; Olfactory Receptor; Sensory Neuron; Antennal Lobe
**Subject Categories** Evolution & Ecology; Methods & Resources; Neuroscience

## Introduction

Sensory regions of the nervous system are, by virtue of their peripheral location and molecularly distinct cell types, particularly amenable for developmental, anatomical, and physiological investigations to obtain a holistic view of the construction and function of neural circuits. Among model sensory systems, the olfactory pathways of *Drosophila melanogaster* are some of the most intensively studied (Benton, 2022; Grabe and Sachse, 2018; Jefferis and Hummel, 2006; Su et al, 2009; Vosshall and Stocker, 2007) (Fig. 1A).

Odor-sensing occurs in two bilaterally symmetric pairs of peripheral organs, the maxillary palps and antennae. These appendages are covered with hundreds of porous sensory hairs, or sensilla, of distinct morphologies (basiconic, trichoid, intermediate, and coeloconic) (Nava Gonzales et al, 2021; Shanbhag et al, 1999, 2000; Shanbhag et al, 1995). Sensilla house the ciliated dendrites of 1–4 olfactory sensory neurons (OSNs), each of which expresses a specific type of odor-binding sensory receptor (or occasionally receptors) that recognize a defined set of volatile chemicals (Couto et al, 2005; de Bruyne et al, 1999; de Bruyne et al, 2001; Fishilevich and Vosshall, 2005; Munch and Galizia, 2016; Silbering et al, 2011). Approximately 25 functional classes of olfactory sensilla on the antenna and maxillary palp can be identified by the stereotypical receptor expression patterns and odor response profiles of the neurons they house (Couto et al, 2005; de Bruyne et al, 1999; de Bruyne et al, 2001; Grabe et al, 2016; van der Goes van Naters and Carlson, 2007; Yao et al, 2005).

Olfactory receptors belong to two families of ligand-gated ion channels: the Odorant receptors (Ors), the founder members of the seven transmembrane domain ion channel (7TMIC) superfamily (Benton and Himmel, 2023; Butterwick et al, 2018; Clyne et al, 1999b; Del Marmol et al, 2021; Gao and Chess, 1999; Himmel et al, 2023; Sato et al, 2008; Vosshall et al, 1999; Wicher et al, 2008), and the Ionotropic receptors (Irs), which are distantly-related to ionotropic glutamate receptors (iGluRs) (Benton et al, 2009). Both Ors and Irs function in known (or presumed) heterotetrameric complexes composed of "tuning" receptor subunits that are thought to directly bind odors, and subunits of one or more broadly expressed co-receptors (Orco for Ors (Larsson et al, 2004); Ir8a, Ir25a, and Ir76b for Irs (Abuin et al, 2011; Vulpe and Menuz, 2021)). Other tuning Ir subunits form hygrosensory and thermosensory receptors with Ir25a and Ir93a co-receptors expressed by sensillar neurons within specialized antennal structures: the sacculus, a three-chambered internal pocket that also houses some olfactory neurons (Ai et al, 2010; Vulpe et al, 2021), and the arista, an elongated cuticular projection (Budelli et al, 2019; Enjin et al,

[1]Center for Integrative Genomics, Faculty of Biology and Medicine, University of Lausanne, CH-1015 Lausanne, Switzerland. [2]Department of Physiology and Neurobiology, University of Connecticut, Storrs, CT 06269, USA. [3]RIKEN Center for Brain Science, Wako, Saitama 351-0198, Japan. [4]Connecticut Institute for Brain and Cognitive Sciences, University of Connecticut, Storrs, CT 06269, USA. [5]These authors contributed equally: Richard Benton, Jérôme Mermet. ✉E-mail: richard.benton@unil.ch; karen.menuz@uconn.edu

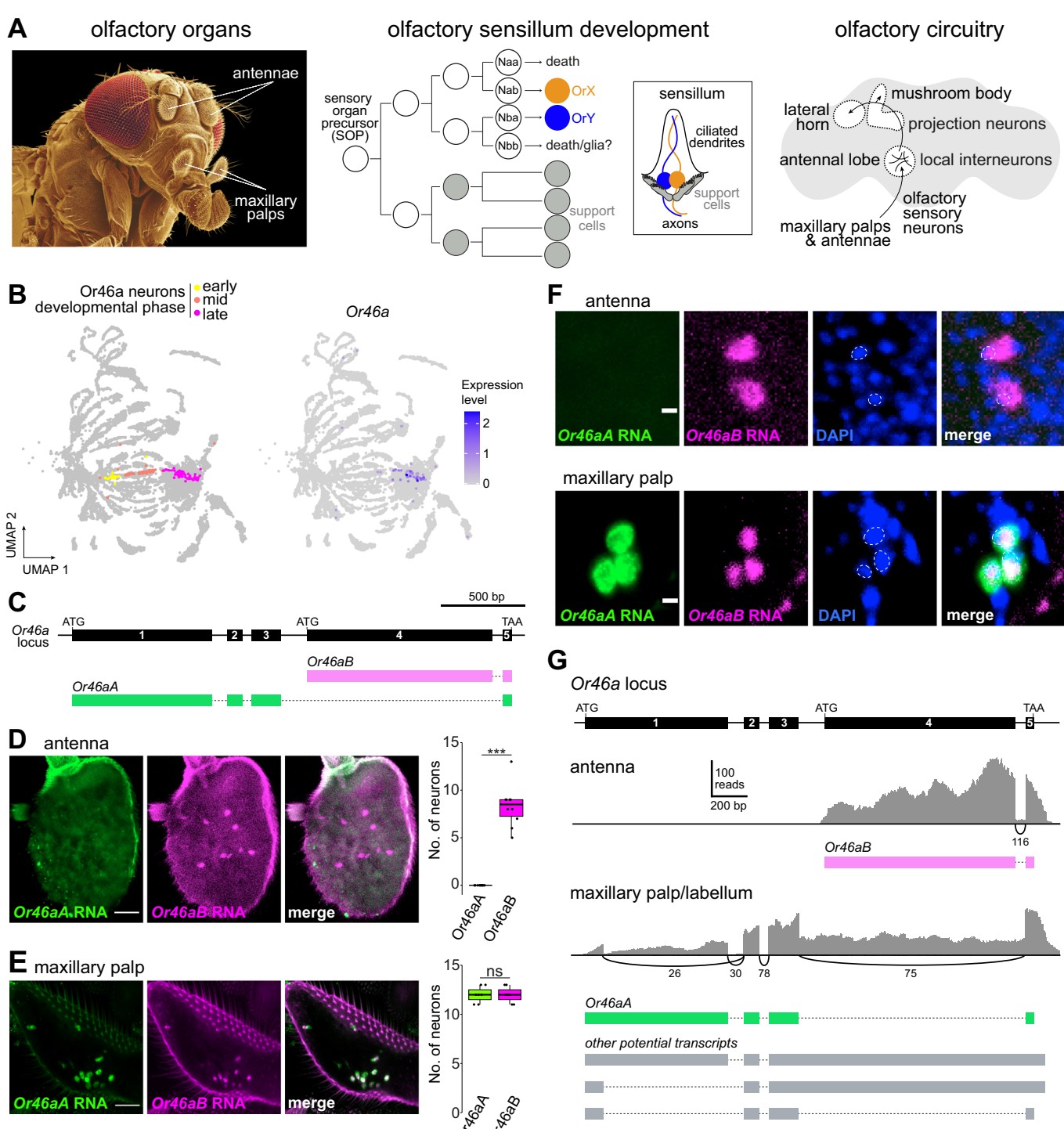

**A** olfactory organs | olfactory sensillum development | olfactory circuitry

**B** Or46a neurons developmental phase: early / mid / late | *Or46a* — Expression level

**C** *Or46a* locus

**D** antenna — *Or46aA* RNA / *Or46aB* RNA / merge — No. of neurons ***

**E** maxillary palp — *Or46aA* RNA / *Or46aB* RNA / merge — No. of neurons ns

**F** antenna — *Or46aA* RNA / *Or46aB* RNA / DAPI / merge; maxillary palp — *Or46aA* RNA / *Or46aB* RNA / DAPI / merge

**G** *Or46a* locus; antenna; maxillary palp/labellum; *Or46aA*; other potential transcripts

2016; Frank et al, 2017; Gallio et al, 2011; Knecht et al, 2017; Knecht et al, 2016; Marin et al, 2020). Finally, a few "Gustatory receptors" (Grs), which are also 7TMICs, function in antennal neurons in $CO_2$ detection (Jones et al, 2007; Kwon et al, 2007) and thermosensation (Ni et al, 2013).

During development, each sensillum derives from an individual sensory organ precursor (SOP) cell in the antennal imaginal disk, which undergoes three stereotyped rounds of division to produce four support cells and four sensory neuron precursors termed Naa,

Nab, Nba and Nbb (Chai et al, 2019; Endo et al, 2007; Endo et al, 2011; Jefferis and Hummel, 2006; Rodrigues and Hummel, 2008) (Fig. 1A). (In coeloconic lineages, the Nbb precursor might differentiate as a glial cell (Endo et al, 2007; Rodrigues and Hummel, 2008; Sen et al, 2005)). Support cells have diverse roles in synthesizing and shaping the sensillar cuticle during development (Ando et al, 2019; Schmidt and Benton, 2020), as well as secreting perireceptor proteins into the sensillar lymph that bathes neuronal dendrites, where they can contribute to sensory responses (Larter

**Figure 1. A new antennal olfactory sensory neuron population.**

(A) Schematic of *D. melanogaster* olfactory system anatomy, development and circuitry (see text for details). Copyright for the false-colored scanning electron micrograph (left): Jürgen Berger/Max Planck Institute for Biology, Tübingen. (B) UMAP of an snRNA-seq atlas of developing antennal neurons colored for developmental phase of the *Or46a* neurons ("early" = 18–30 h after puparium formation (APF), "mid" = 36–48 h APF, "late" = 56–80 h APF) (left) and expression of *Or46a* transcripts (right). Data from (preprint: Mermet et al, 2025). Gene expression levels, here and in other UMAPs, are residuals from a regularized negative binomial regression and have arbitrary units. (C) Structure of the *Or46a* locus and the transcript isoforms for *Or46aA* and *Or46aB*. (D, E) RNA FISH with isoform-specific probes for *Or46aA* and *Or46aB* in a whole-mount antenna (D) and maxillary palp (E). Scale bars, 25 μm. Quantifications of neuron numbers are shown on the right. Box plots show median (thick line) and first and third quartiles, while whiskers indicate data distribution limits, overlaid with individual data points (the topmost point for Or46aB neurons is an outlier) (n = 10 antennae (D) and 7 maxillary palps (E)). ***$P$ = 7e$^{-7}$; ns, $P$ = 1, $t$ test. (F) High-magnification images of RNA FISH for *Or46aA* and *Or46aB* in an antenna and a maxillary palp. Dashed lines outline the nuclei (stained with DAPI), revealing greater nuclear sequestration of *Or46aB* in the maxillary palp neurons compared to *Or46aA* transcripts, or to *Or46aB* transcripts in the antenna. Scale bars, 3 μm. (G) *Or46a* isoform expression analyzed from bulk RNA-seq data of antennal and maxillary palp/labellar tissue (Bontonou et al, 2024; Data ref: Bontonou et al, 2024). Top: structure of the *Or46a* locus. Sashimi plots generated with IGV (Thorvaldsdottir et al, 2013) showing mapped reads (gray) from the indicated tissue transcriptomes aligned to the *Or46a* locus. Quantifications of splice junction mapping reads are indicated beneath the plots, and the predicted transcript isoforms in each tissue are shown below (*Or46aB* in magenta, *Or46aA* in green). Potential transcripts in the palp shown in gray are unlikely to encode functional receptor proteins (see "Results").

et al, 2016; Sun et al, 2018; Xu et al, 2005). Sensory neuron precursors are thought to express unique combinations of transcription factors that, together with asymmetric Notch activity between daughter cells of each division, result in unique terminal identities of the olfactory neurons (Barish and Volkan, 2015; Chai et al, 2019; Endo et al, 2007; Endo et al, 2011; preprint: Mermet et al, 2025). In most sensillar classes, one or more sensory neuron precursors stereotypically undergo programmed cell death, leaving fewer than four functional neurons in mature sensilla (Endo et al, 2007; Endo et al, 2011; Prieto-Godino et al, 2020; Sen et al, 2004).

Each population of sensory neurons expressing the same receptor(s) innervates a specific glomerulus in the antennal lobe, the initial processing center in the brain (Couto et al, 2005; Fishilevich and Vosshall, 2005; Gao et al, 2000; Silbering et al, 2011; Vosshall et al, 2000). Here these sensory neurons synapse with local neurons (LNs), which mediate interglomerular interactions (Chou et al, 2010; Wilson, 2013) and projection neurons (PNs), which transmit sensory information to higher processing centers, the mushroom body and lateral horn (Bates et al, 2020; Marin et al, 2020; Marin et al, 2002; Schlegel et al, 2021; Wong et al, 2002) (Fig. 1A).

The global view of the organization and function of the *D. melanogaster* olfactory system has emerged from diverse experimental approaches over the past 30 years. Odor response profiles of nearly all receptors and/or sensory neurons have been obtained through measurement of odor-evoked activity in vivo by extracellular electrophysiological recordings from individual sensilla (e.g., (de Bruyne et al, 1999; de Bruyne et al, 2001; Hallem and Carlson, 2006; Yao et al, 2005)), optical imaging of activity in sensory neuron axonal termini in glomeruli (e.g., (Silbering et al, 2011; Wang et al, 2003)) and/or through characterization of receptors in heterologous expression systems (e.g., (Ruel et al, 2021; Sato et al, 2008)). In situ analysis of the expression of endogenous receptors or transgenic promoter reporters (Benton et al, 2009; Couto et al, 2005; Fishilevich and Vosshall, 2005; Grabe et al, 2016; Silbering et al, 2011) has been complemented with comprehensive, high resolution transcriptomic analyses of OSNs and PNs (Arguello et al, 2021; Li et al, 2017, 2020; McLaughlin et al, 2021). Receptor promoter transgenic reporters have also enabled neuronal tracing to produce a near-complete, neuron-to-glomerulus map (Couto et al, 2005; Fishilevich and Vosshall, 2005; Silbering et al, 2011), which has recently been greatly extended by electron microscopic (EM) analyses that also offer insights into the glomerular microcircuitry of sensory neurons, LNs and PNs (Bates

et al, 2020; Marin et al, 2020; Rybak et al, 2016; Schlegel et al, 2021; Tobin et al, 2017), as well as the innervations of PNs in higher brain regions (Bates et al, 2020; Jefferis et al, 2007; Marin et al, 2020; Schlegel et al, 2021). Insights into how this circuitry forms have been discovered through a wealth of forward and reverse molecular genetic investigations of OSN and PN development (Barish and Volkan, 2015; Brochtrup and Hummel, 2011; Hong and Luo, 2014; Jefferis and Hummel, 2006). The behavioral role(s) of many individual sensory pathways have been revealed by genetic manipulations of receptors, as well as artificial inhibition or activation of the neurons in which they are expressed (e.g., (Ai et al, 2010; Stensmyr et al, 2012; Suh et al, 2004; Tumkaya et al, 2022; Wu et al, 2022)). Finally, comparative analysis of the *D. melanogaster* olfactory system with that of other drosophilids and more distantly-related insect species has begun to uncover how individual sensory pathways diverge structurally and/or functionally during evolution (Auer et al, 2020; Dekker et al, 2006; Depetris-Chauvin et al, 2023; Hansson and Stensmyr, 2011; Prieto-Godino et al, 2016; Prieto-Godino et al, 2017; Ramdya and Benton, 2010; Takagi et al, 2024; Zhao and McBride, 2020).

These numerous investigations into *D. melanogaster*'s olfactory pathways provide essential resources for the field. However, integration of information across different studies can be difficult due to conflicting assignments of some receptors to neuron types and sensillar classes, inconsistent naming of antennal lobe glomeruli, and ongoing updates to the olfactory map. In this work, we first "complete" this map through the discovery of a previously undescribed antennal OSN type, which resolves long-known inconsistencies in sensillar identification. We also reveal a neuron that relies on both Ir and Or tuning receptors, the only such "hybrid" olfactory neuron characterised in *D. melanogaster*. These findings spurred us to compile an integrated data resource to overcome the dispersal of pertinent information with disparate anatomical and molecular naming across the literature. We also created updated representations of both the complete sensillar classes and the antennal lobe glomeruli to serve as standardized references for the field.

## Results and discussion

### A novel antennal Or sensory neuron type

Within a single-nuclear RNA-sequencing (snRNA-seq) atlas of the developing antenna (preprint: Mermet et al, 2025), we observed a

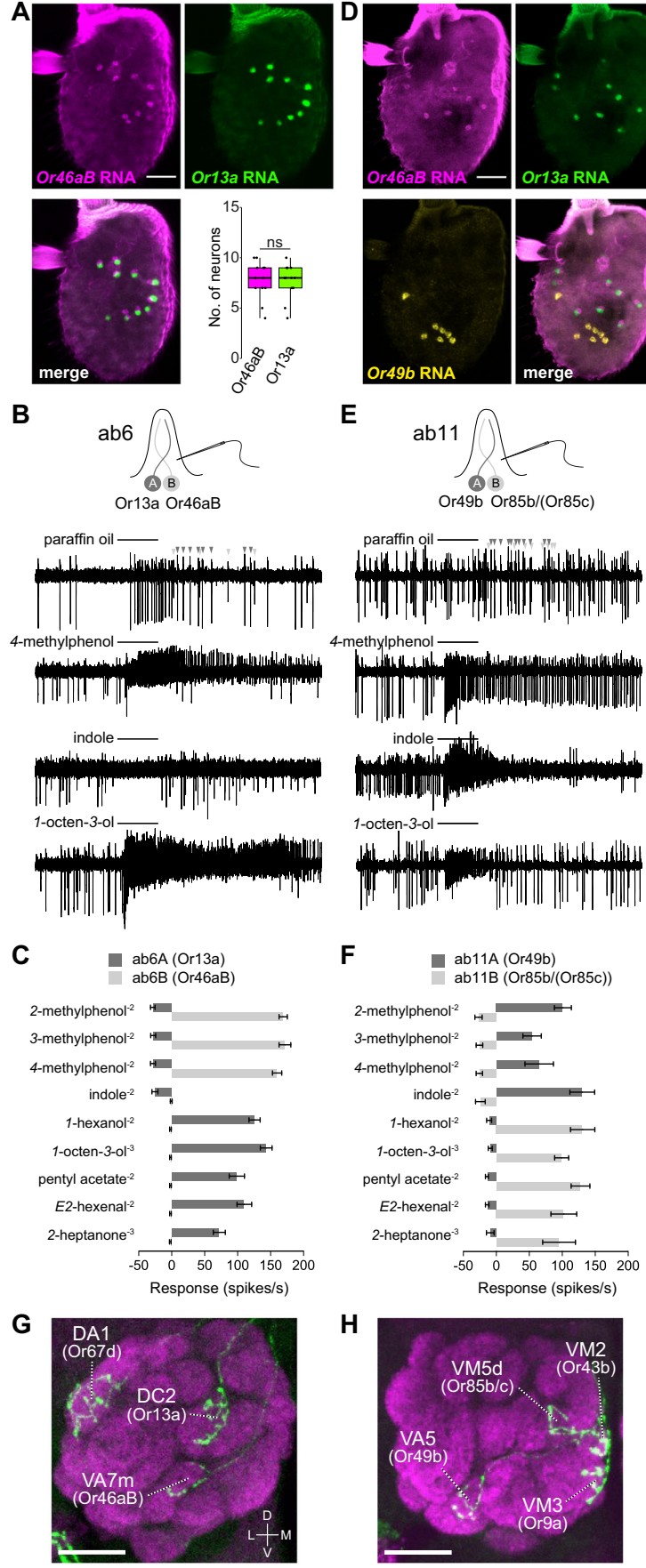

◄ **Figure 2. Molecular, functional, and anatomical validation of ab6 and ab11 sensilla.**

(A) RNA FISH on a whole-mount antenna illustrating the pairing of Or46aB and Or13a neurons. Quantification of neuron numbers is shown on the right ($n = 12$ antennae). Box plots show median (thick line) and first and third quartiles, while whiskers indicate data distribution limits, overlaid with individual data points. ns, $P = 0.91$, $t$ test. Scale bar, 25 μm. (B) Representative traces of single-sensillum recordings of GFP$^+$ ab6 sensilla from Or13a > mCD8:GFP flies illustrating neuronal responses to the indicated odors (0.5 s stimulation time, black bars). In the top trace, two spike amplitudes, reflecting distinct neurons, are highlighted with dark and light gray arrowheads. (C) Quantification of odor-evoked responses in ab6A (large spiking) and ab6B (small spiking) neurons. Odor dilutions (v/v in paraffin oil) are shown in superscript. Solvent-corrected responses (mean ± SEM) from $n = 6-7$ sensilla, biological replicates, are shown. See Source Data for spike counts. (D) RNA FISH on a whole-mount antenna illustrating the distinct distribution of Or49b neurons compared to the paired Or46aB and Or13a neurons. Scale bar, 25 μm. (E, F) As in (B, C), but for recordings of GFP$^+$ ab11 sensilla ($n = 6-8$ sensilla) from Or49b > mCD8:GFP flies. (G) Antennal lobe projections of clonally-marked OSNs visualized with GFP immunofluorescence (green) together with nc82 neuropil stain (magenta) revealing co-labeling of neurons innervating DC2 (Or13a) and VA7m (inferred to be Or46aB) glomeruli. Data were re-analyzed and re-processed from a dataset generated in (Endo et al, 2007); of 12 brains with DC2-labeled neurons, all had VA7m-labeled neurons (1 with weak labeling), strongly supporting the innervation patterns of the paired neurons in ab6. In this image, DA1 (Or67d) OSNs are also labeled, representing an independent clone in the at1 lineage. Scale bar, 20 μm. (H) Antennal lobe projections of clonally-marked OSNs innervating VA5 (Or49b) and VM5d (Or85b/(Or85c)) glomeruli. Data were re-analyzed and re-processed from a dataset generated in (Endo et al, 2007); of four brains with VA5-labeled neurons, three also had VM5d-labeled neurons, supporting the pairing of these neurons in ab11. In this image, VM2 (Or43b) and VM3 (Or9a) OSNs are also labeled, representing an independent clone in the ab8 lineage. Scale bar, 20 μm. Source data are available online for this figure.

cell cluster expressing *Or46a* (Fig. 1B). Transcripts for this gene had previously been detected by RT-PCR and in bulk RNA-seq datasets of the antenna (Clyne et al, 1999a; Menuz et al, 2014), but never assigned to a specific cell type. The *Or46a* locus encodes two receptors, Or46aA and Or46aB, which share the same C-terminus encoded by a common last exon (Fig. 1C). Through RNA fluorescence in situ hybridization (FISH) with isoform-specific probes, we detected expression of transcripts for *Or46aB* in ~8 neurons in the antenna, but not *Or46aA* (Fig. 1D). As a control, we performed RNA FISH on maxillary palps, verifying that both *Or46a* probes detect the same neurons in this organ, as described previously (Ray et al, 2007) (Fig. 1E). However, we observed that the signals of the two probes were spatially distinct (Fig. 1F): *Or46aA* was detected both in the cytoplasm and the nucleus, while *Or46aB* appeared predominantly nuclear in palp OSNs, despite being readily detected in the cytoplasm of antennal OSNs (Fig. 1F). This phenomenon is reminiscent of the nuclear retention of transcripts of downstream genes in tandem clusters of *Or*s in ants (Brahma et al, 2023).

To understand the reason for this differential location, we assessed transcripts arising from the *Or46a* locus in antenna and maxillary palp/labellum bulk transcriptomes (Bontonou et al, 2024; Data ref: Bontonou et al, 2024) (Fig. 1G). In the antenna, we detected transcripts only for *Or46aB*, as expected. In the maxillary palp/labellum transcriptome, we detected several alternative splicing events; many of these correspond to splicing events in *Or46aA*, as previously characterized by RT-PCR of full-length transcripts (Ray et al, 2007). Importantly, although we found transcripts including *Or46aB* exons, we did not find any evidence for proper splicing between exons 4 and 5. This lack of splicing means that all transcripts with *Or46aB* exons contain a frameshift that renders exon 5 unable to encode for the essential ion channel pore region. We also observed sequences with an unusual alternative splicing event in the first exon of *Or46aA* that would prevent them from encoding a functional receptor. We suggest that many or all of these transcripts are aberrant splice variants initiating from the *Or46aA* promoter and likely fail to be exported efficiently from the nucleus or are rapidly degraded in the cytoplasm. The simplest interpretation of these data is that antennal neurons only express Or46aB protein, while maxillary palp neurons predominantly or only express Or46aA.

## "Completing" the olfactory map in the antenna and antennal lobe

We next sought the antennal sensillum class in which the newly-identified Or46aB neurons are housed, taking advantage of odor-to-neuron-to-sensillum maps defined by electrophysiological and histological analyses (Couto et al, 2005; de Bruyne et al, 2001; Grabe et al, 2016) and knowledge that Or46aB responds to methylphenols when expressed in heterologous neurons (Ray et al, 2014). We predicted that Or46aB is expressed in the antennal basiconic 6 (ab6) sensillar class "B" neuron (i.e., with the smaller spike amplitude) as this ab6B neuron responds strongly and selectively to methylphenols (de Bruyne et al, 2001; Hallem et al, 2004). The molecular identity of the ab6A neuron (i.e., with the larger spike amplitude) has been inconsistently described in the literature (see "Terminology" section in the Methods), but the best evidence is that this neuron class expresses Or13a, due to the similar odor-tuning profiles of ab6A neurons measured by single-sensillum recordings (de Bruyne et al, 2001) and Or13a neurons measured by calcium imaging (Galizia et al, 2010).

We tested this prediction through two-color RNA FISH using probes against these receptors, observing precise pairing of Or46aB and Or13a neurons (Fig. 2A). We further investigated the neuronal composition and function of this sensillum through targeted electrophysiological recordings of sensilla labeled with GFP driven by *Or13a-Gal4*. Observation of basal spiking patterns confirmed the presence of two neurons, based upon their distinct spike amplitudes (Fig. 2B), countering a previous claim that these sensilla house a single neuron (Lin and Potter, 2015). Profiling of the odor-evoked responses confirmed that the A neuron responds most strongly to *1*-octen-*3*-ol and robustly to *1*-hexanol, *E2*-hexenal, pentyl acetate and *2*-heptanone, matching the profile of ab6A neurons previously defined by electrophysiological recordings (de Bruyne et al, 2001) and of Or13a neurons measured with calcium imaging (Galizia et al, 2010). As previously described for ab6B neurons (de Bruyne et al, 2001; Hallem et al, 2004), the neuron paired with Or13a neurons responds to methylphenols (Fig. 2B,C), matching the response profile of heterologously-expressed Or46aB (Ray et al, 2014). Together, these data support the proposal that Or13a and Or46aB are expressed in the originally-defined ab6 sensillum class (de Bruyne et al, 2001).

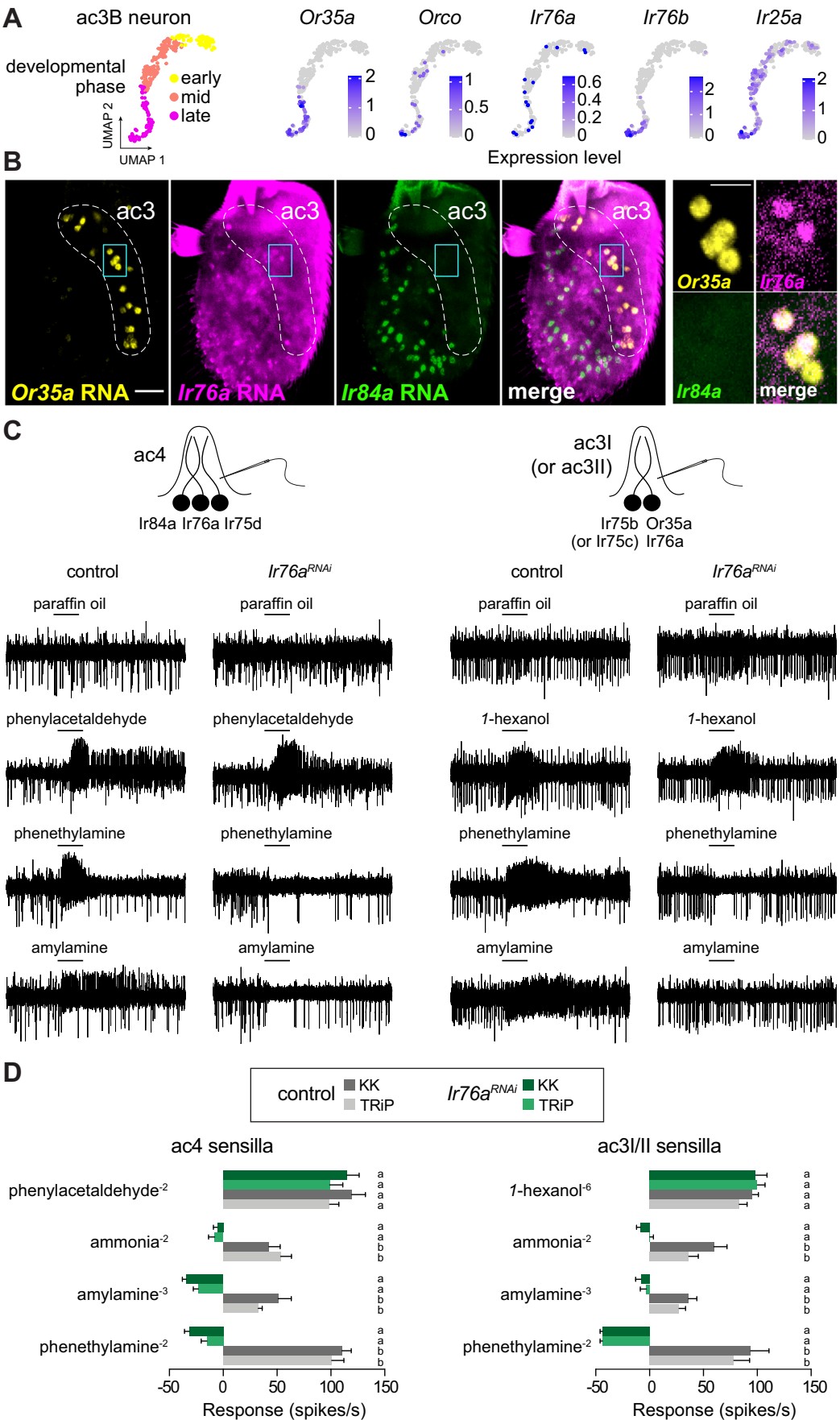

◄ **Figure 3. A hybrid Or/Ir OSN population.**

(A) Top: UMAPs of the ac3B neurons at different developmental phases extracted from the snRNA-seq atlas (Fig. 1B) (preprint: Mermet et al, 2025) illustrating the expression patterns of the indicated receptor genes. (B) RNA FISH on a whole-mount antenna of control ($w^{1118}$) animals with probes targeting the indicated transcripts. The ac3 sensilla zone is indicated; it is distinct from the ac4 zone where Ir84a neurons (and most Ir76a neurons) are located. Scale bar, 25 µm. Right: ac3B neurons co-expressing Or35a and Ir76a (but not paired with ac4 Ir84a-expressing neurons) in a single confocal Z-slice. Scale bar, 10 µm. (C) Representative traces of single-sensillum recordings from ac4 and ac3 sensilla in control and Ir76a$^{RNAi}$ flies (TRiP line) illustrating neuronal responses to the indicated odors (0.5 s stimulation time, black bars). We note that ac3I (housing Ir75b neurons) and ac3II (housing Ir75c neurons) subtypes cannot be distinguished electrophysiologically in *D. melanogaster*, so they were considered collectively in these recordings. (D) Electrophysiological responses to the indicated ligands in ac4 and ac3 sensilla from antennae of two independent lines of control and Ir76a$^{RNAi}$ animals. Solvent-corrected responses (mean ± SEM) of the combined activities of all neurons in each sensillum are shown ($n = 5$–7 ac4 sensilla and 6–13 ac3 sensilla). Letters a and b indicate significant differences ($P < 0.05$) between genotypes for a given odorant (two-way ANOVA followed by Tukey's multiple comparisons test to examine simple effects by odorant). See Source Data for spike counts and details on statistical analyses, including precise $P$ values. Source data are available online for this figure.

One complication with this assignment is that ab6B has previously been posited to express Or49b (e.g., (Couto et al, 2005; Grabe et al, 2016; Hallem et al, 2004)), likely because this receptor also responds to methylphenols (Hallem et al, 2004). Although it is possible that Or49b and Or46aB are co-expressed in ab6B, there is no evidence for this in our snRNA-seq datasets (preprint: Mermet et al, 2025). Moreover, we recently demonstrated using RNA FISH that Or49b neurons are paired with those expressing Or85b/(Or85c) (in this study, we place receptors in parentheses if their function is unclear) (Takagi et al, 2024). The simplest interpretation is that there are two discrete classes of sensilla, one with Or13a and Or46aB neurons and the other with Or85b/(Or85c) and Or49b neurons. These classes may have been conflated previously due to the common sensitivity of both Or46aB and Or49b to methylphenols.

To validate that Or49b and Or85b/(Or85c) define a unique sensillum class, we first used FISH to verify that Or49b neurons are not paired with Or13a neurons or Or46aB neurons in the antenna (Fig. 2D). We next used Or49b-Gal4 to mark these sensilla with GFP and performed electrophysiological recordings with the same set of odors as above (Fig. 2E,F). As expected, we found that the response profile of sensilla housing Or49b and Or85b/(Or85c) neurons is similar to those containing Or13a and Or46aB neurons. However, two key features indicate that the sensilla are distinct. First, methylphenols activate the A neuron in Or49b sensilla (Fig. 2E,F), but the B neuron in Or13a sensilla (Fig. 2B,C), while odors such as 2-heptanone and 1-octen-3-ol activate the B neuron in Or49b sensilla, but the A neuron in Or13a sensilla. Second, the responses of Or13a and Or49b sensilla to indole, an odor reported to strongly activate Or49b (Ruel et al, 2021) differ: the A neuron in Or49b sensilla responds robustly to this odor, whereas neurons in Or13a sensilla do not (Fig. 2B,C,E,F), as originally reported in ab6 (de Bruyne et al, 2001). Together, the data confirm that these receptors are expressed in two separate classes of sensilla, and that the ab6 sensilla response profile is matched best by the sensillum housing Or13a and Or46aB neurons. We propose to name the sensillum housing Or49b and Or85b/(Or85c) neurons ab11 (see the "Terminology" section in "Methods").

We next sought where Or46aB antennal OSNs project in the brain. Functional transgenic drivers for the Or46aB neuron have been difficult to generate (Couto et al, 2005; preprint: Tirian and Dickson, 2017), likely reflecting the unusual genomic organization of this locus (Fig. 1C). This unfortunately prevents direct visualization of their glomerular target in the antennal lobe. However, we hypothesized that these neurons innervate the VA7m

glomerulus. Three pieces of evidence support this possibility: VA7m is the last "orphan" glomerulus in the antennal lobe (Schlegel et al, 2021), i.e., without molecularly defined sensory innervations. Second, the glomerulus is adjacent to the VA7l glomerulus, which is innervated by maxillary palp Or46aA neurons (Couto et al, 2005). Such an assignment aligns with evidence that evolutionarily closely-related receptors tend to be expressed in neurons that project to nearby glomeruli (Couto et al, 2005; Silbering et al, 2011). Most compellingly, clonal labeling of OSNs demonstrated that the sister neuron of Or13a—i.e., arising from the same SOP lineage, which we have now established is the Or46aB neuron (Fig. 2A–C)—innervates VA7m (Fig. 2G) (Endo et al, 2007). This neuron-to-glomerulus assignment effectively completes the antennal lobe map. In addition, while re-analyzing data from (Endo et al, 2007), we found several examples of brains in which VA5 (Or49b) neurons are co-labeled with VM5d (Or85b/(Or85c)) neurons, supporting the pairing of these neurons in ab11 (Fig. 2H). This co-labeling was previously overlooked as VM5d (Or85b/(Or85c)) neurons were mostly co-labeled with DM2 (Or22a/(Or22b)) neurons, corresponding to the co-housing of these OSN types in ab3.

## A "hybrid" olfactory pathway expressing a functional Or- and Ir-tuning receptor

Our snRNA-seq atlas (preprint: Mermet et al, 2025) revealed a second, previously-unreported expression pattern: weak expression of Ir76a in Or35a-expressing cells that correspond to the B neurons in antennal coeloconic 3 (ac3) sensilla (Fig. 3A). (Stronger Ir76a expression was detected in the ac4 Ir76a neuron (Benton et al, 2009; preprint: Mermet et al, 2025)). We confirmed these transcriptomic data in vivo using RNA FISH, which detected Ir76a transcripts in several, though not all, Or35a ac3B neurons (Fig. 3B).

The expression of Ir76a in ac3B was intriguing because while most odor responses of the broadly-tuned ac3B neuron depend upon Ors (Silbering et al, 2011; Yao et al, 2005), responses to amines – notably phenethylamine and amylamine—require instead the Ir co-receptors Ir25a and Ir76b (Vulpe and Menuz, 2021), which are also expressed in these cells (Fig. 3A) (Task et al, 2022). As these amines are amongst the best agonists of ac4 Ir76a neurons (Silbering et al, 2011), we hypothesized that Ir76a is the tuning receptor mediating amine responses in ac3B neurons. We tested this possibility through single-sensillum electrophysiological analyses of existing Ir76a$^{RNAi}$ flies because an Ir76a null mutant, the optimal reagent, is not yet available (Fig. 3C,D). Using two

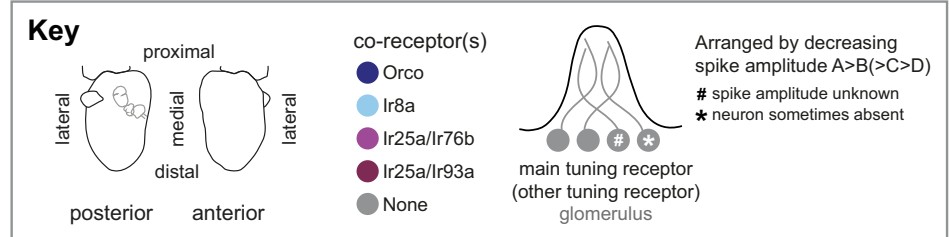

**basiconics (maxillary palp)**

pb1 — Or42a, Or71a — VM7d, VC2

pb2 — Or85e Or33c, Or46aA — VC1, VA7l

pb3 — Or59c, Or85d — VM7v, VA4

**large basiconics**

ab1 — Or42b, Or92a, Gr21a Gr63a, Or10a (Gr10a) — DM1, VA2, V, DL1

ab2 — Or59b, Or85a (Or33b) — DM4, DM5

ab3 — Or22a (Or22b), Or85b (Or85c) — DM2, VM5d

**thin basiconic**

ab4 — Or7a, Or56a (Or33a) — DL5, DA2

**small basiconics**

ab5 — Or82a, Or47a (Or33b) — VA6, DM3

ab6 — Or13a, Or46aB — DC2, VA7m

ab7 — Or98a, Or67c — VM5v, VC4

ab8 — Or43b, Or9a — VM2, VM3

ab9 — Or69aA/B, Or67b — D, VA3

ab10 — Or67a, Or49a Or85f — DM6, DL4

ab11 — Or49b, Or85b (Or85c) — VA5, VM5d

**intermediates**

ai2 — Or83c, Or23a — DC3, DA3

ai3 — Or19a (Or19b), Or2a, Or43a — DC1, DA4m, DA4l

**coeloconics**

ac1 — Amt, Ir31a, Ir92a, Ir75d — VM6, VL2p, VM1, VL1

ac2 — Ir75a, Ir41a, Ir75d — DP1l, VC5, VL1

ac3I — Ir75b, Or35a Ir76a — DL2d, VC3

ac3II — Ir75c, Or35a Ir76a — DL2v, VC3

ac4 — Ir84a, Ir76a (Or35a), Ir75d (Or35a), Ir75d — VL2a, VM4, VL1

**trichoids**

at1 — Or67d — DA1

at4 — Or47b, Or65a (Or65b) (Or65c), Or88a — VA1v, DL3, VA1d

**sacculus**

sacI — Ir21a, Ir40a, Ir68a — VP1l, VP4, VP1m

sacII — Ir68a, Ir40a, Ir40a — VP5, VP4, VP1d

sacIII-v — Ir64a, Amt (Or35a) — DC4, VM6

sacIII-d — Ir64a, Amt (Or35a) — DP1m, VM6

**arista**

Ir21a, Gr28b.d — VP3, VP2

**Figure 4. Antennal and maxillary palp sensory sensilla organization.**

Updated neuronal composition of all sensillar classes in the maxillary palp and antenna, including tuning receptors, co-receptors, and the corresponding glomerular targets in the antennal lobe. Tuning receptors shown in parentheses are reported to be expressed in the neuron population but have not yet been shown to contribute to their odor responses; in some cases, these might be non-functional. In ab10 and at4, a specific neuron is sometimes lacking in mature sensilla (asterisks), likely due to promiscuous programmed cell death (preprint: Mermet et al, 2025; Nava Gonzales et al, 2021); the frequency of absence is strain-dependent (preprint: Mermet et al, 2025). The approximate distribution of olfactory sensilla within the sensory organs (shown above each sensillum) is adapted from (Grabe et al, 2016) except for ab3 and ab11, which were mapped using image data from (Takagi et al, 2024), and ac3I and ac3II, which were mapped using data from (Mika et al, 2021). While the overall distribution is stereotyped between antennae, there is variation in the individual position of sensilla. The anterior/posterior distribution of large basiconic sensilla does not fully agree with an earlier mapping (de Bruyne et al, 2001), which might reflect differences between studies in the definition of the anterior and posterior surfaces.

independent transgenic RNAi lines, covering non-overlapping regions of the coding sequence, we first verified the efficiency of *Ir76a^{RNAi}* in ac4 sensilla, observing complete loss of responses to amine ligands of Ir76a neurons, while responses of the co-housed Ir84a neurons to phenylacetaldehyde were unchanged (Fig. 3C,D). In ac3B neurons, amine responses were similarly abolished by both *Ir76a^{RNAi}* lines, while responses to the Or35a/Orco-dependent ligand *1*-hexanol were unaffected (Fig. 3C,D).

These results indicate that the ac3B neuron is, to our knowledge, the first unambiguous example of an OSN expressing functionally relevant combinations of tuning and co-receptors of both Or and Ir families. Interestingly, recent snRNA-seq and RNA FISH in the mosquito *Aedes aegypti* identified a few OSN populations in the antenna and maxillary palp expressing putatively complete sets of both Or and Ir complexes (preprint: Adavi et al, 2024; Herre et al, 2022), indicating that similar "hybrid" neuron types might exist in other species.

## A new integrated dataset of the developmental, anatomical, and functional properties of the *D. melanogaster* olfactory system

Our discoveries of the Or46aB and hybrid Or35a/Ir76a sensory channels both highlighted prior inaccuracies and omissions in the antennal and antennal lobe maps and exemplified the power of using information from disparate sources to extract new insights. We therefore reasoned that it was timely to systematically integrate current data resources on diverse developmental, anatomical, and functional properties of the olfactory and hygro/thermosensory systems. Building on a foundational data resource generated nearly a decade ago (Grabe et al, 2016) and from several recent studies on hygrosensors and thermosensors (Budelli et al, 2019; Enjin et al, 2016; Frank et al, 2017; Gallio et al, 2011; Knecht et al, 2017; Knecht et al, 2016; Marin et al, 2020), we made substantial new additions and corrections regarding receptor expression patterns, as well as neuronal and sensillar annotations. For example, in addition to the definition of ab6 and ab11 described above, we distinguish the classes of antennal intermediate (ai2, ai3) and trichoid (at1, at4) sensilla more clearly, as these have been conflated in the past (e.g., in (Couto et al, 2005) ai2 and ai3 sensilla were referred to as "at2" and "at3", respectively). We also update the definition of ac3 sensilla that comprise two subtypes, ac3I and ac3II, housing Ir75b and Ir75c neurons respectively (Prieto-Godino et al, 2017), each together with the Or35a/Ir76a neurons characterized here.

We also collated improved quantitative estimates of neuronal populations, favoring numbers from analyses of in situ gene expression—including many new quantifications using HCR FISH (Fig. EV1), other numbers from the literature (e.g., (preprint:

Mermet et al, 2025)) and from very recent EM connectomic datasets (Dorkenwald et al, 2024; Schlegel et al, 2021; Schlegel et al, 2024) – rather than transgenic reporters as in (Grabe et al, 2016), which do not always faithfully reflect endogenous gene expression. We additionally integrated several developmental properties, such as expression of proneural and other fate determinants, as well as available anatomical information on LNs (Chou et al, 2010) and uniglomerular PNs (Schlegel et al, 2021). Finally, we incorporated comparative datasets of OSN numbers and glomerular size available for several species in the *Drosophila* group (Depetris-Chauvin et al, 2023).

Behavior is of course the *raison d'être* of the olfactory system, and there is a wealth of information on the contributions of many individual olfactory pathways (e.g., (Badel et al, 2016; Semmelhack and Wang, 2009; Wu et al, 2022)). For certain sensory channels, such as those detecting pheromones, several studies provide consistent evidence for their behavioral role(s) (Kurtovic et al, 2007; Taisz et al, 2023). However, for the majority of pathways, their contribution to odor-evoked behaviors—as assessed by loss-of-function or artificial neuronal activation approaches—are highly context-dependent (Currier and Nagel, 2020), influenced by the experimental assay design (Chin et al, 2018; Tumkaya et al, 2022; Wu et al, 2022), environmental conditions (e.g., air currents (Bell and Wilson, 2016; Matheson et al, 2022; Stupski and van Breugel, 2024)), other simultaneous olfactory and taste inputs (Grabe and Sachse, 2018; Oh et al, 2021; Wilson, 2013) and the internal state of the fly (e.g., starvation (Ko et al, 2015; Lebreton et al, 2015; Root et al, 2011)). Collectively, these studies support the idea that many sensory channels function as part of a "combinatorial code" to control behavioral outputs. We have therefore adopted the more general idea of the "sensory scene" within which a particular olfactory pathway might function (Schlegel et al, 2021). This classification is largely defined by the likely ecological source of the odor(s) to which a given OSN responds (Mansourian and Stensmyr, 2015). We caution that such classification is tentative, as some chemicals can be found in many different biological settings.

The full integrated dataset is provided in Dataset EV1; this is also available online (https://shorturl.at/gznii), with the aim that such a dataset can be supplemented with information emerging in future investigations, such as additional molecular markers (McLaughlin et al, 2021; preprint: Mermet et al, 2025; Xie et al, 2021), functional properties of individual sensory pathways, and further data from other species of drosophilids (Depetris-Chauvin et al, 2023). Accompanying this resource, we have created schematics highlighting some key organizational properties of sensory sensilla (Fig. 4). We have also generated labeled atlases and movies depicting coronal (anterior-to-posterior) (Fig. 5; Dataset EV2) and transverse (dorsal-to-ventral) (Fig. EV2; Dataset EV2) sections through the antennal lobe based on 3D

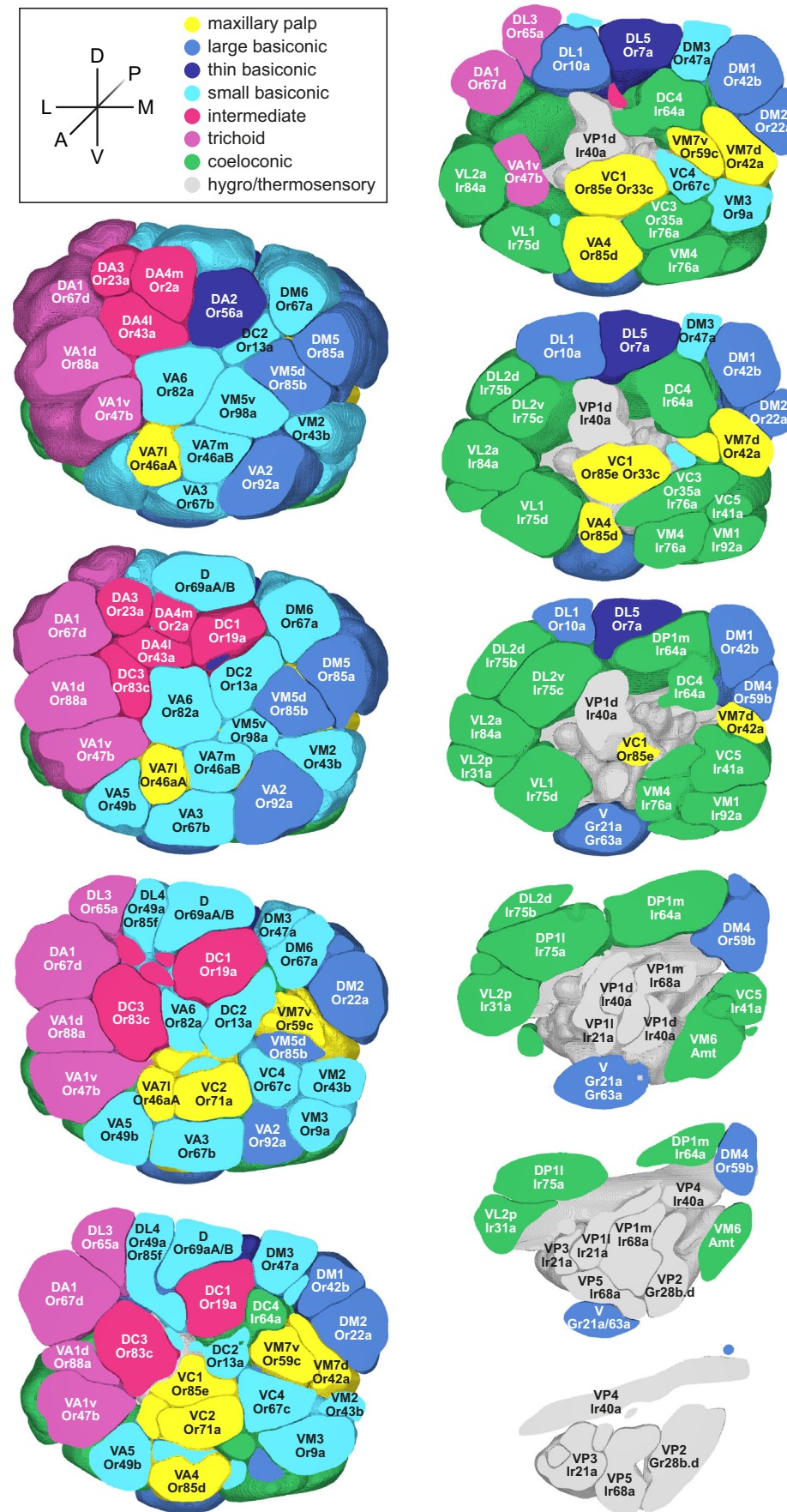

**Figure 5. Antennal lobe atlas of coronal sections.**

Coronal sections through an updated antennal lobe atlas adapted from glomerular meshes based on the female adult fly brain (FAFB) EM dataset (Bates et al, 2020) (see "Methods" and Dataset EV2). Anterior is top-left, and posterior is bottom-right. The atlas contains updated tuning receptor and glomerular names (Schlegel et al, 2021), and glomeruli are color-coded by sensillar class. Glomeruli innervated by OSNs from sacculus chamber III are colored green, as they are most similar to coeloconic neurons. For compactness, only the main known tuning receptor is indicated. For an alternative set of transverse sections along the dorsal-ventral axis, see Fig. EV2. See Dataset EV2 for interactive and modifiable versions as well as finer-grained coronal and transverse movies of sections through the antennal lobe.

glomerular meshes from a recent EM-based atlas (Bates et al, 2020). Together, these should serve as practical guides during, for example, anatomical and neurophysiological investigations.

## Illustration of insights from the integrated dataset

While the information compiled above should serve as a useful reference source during the study of specific sensory pathways, we describe in this section a few examples of insights that can be gleaned from global analyses using these updated data.

*Relationship between OSN precursor identity and OSN morphology:* unlike the odor response profile, OSN spike amplitude is not defined by the tuning receptor (Hallem et al, 2004) but rather reflects the morphology of the corresponding OSN. OSNs with greater dendritic surface area, typically due to extensive branching of the sensory cilia endings, have larger spike amplitudes (Nava Gonzales et al, 2021; Shanbhag et al, 1999, 2000). Essentially all sensilla house neurons of distinct, stereotyped spike amplitudes, implying a hard-wired genetic control of neuronal morphology. We asked whether these differences reflect the corresponding neuronal precursor identity. By examining sensilla with two OSNs, we found that the neurons with larger spike amplitudes (A neurons) and those with smaller spike amplitudes (B neurons) were derived from a similar proportion of Nab and Nba precursors (Fig. 6A; Dataset EV1). Similarly, in 3-OSN sensilla the A neuron was derived either from Nab (at4, ac2, ac4) or Nba (ai3), and in 4-OSN sensilla the A neuron was derived from either Nba (ab1) or Nbb (ac1). These observations indicate that OSN morphology is not simply derived from the developmental pathway characteristic of different OSN precursors, such as the Notch status after asymmetric cell division (Endo et al, 2007; Endo et al, 2011). Extraction of transcripts enriched in large or small spiking neurons from snRNA-seq datasets (Li et al, 2020; McLaughlin et al, 2021; preprint: Mermet et al, 2025) might reveal candidate molecules underlying differences in cilia morphology, an outstanding question in sensory biology in insects and other animals (Maurya, 2022).

*Sexual dimorphisms and species differences in OSN numbers:* many insects have sex-specific olfactory pathways, most famously in moths that possess male-specific populations detecting female pheromones (Nakagawa et al, 2005). By contrast, in *D. melanogaster* sexual dimorphisms in the size of OSN populations appear to be limited. With our revised set of neuron numbers (Dataset EV1), we re-visited this issue by plotting the female:male ratio of OSN numbers, where data are available. While we confirmed that sexual dimorphisms are modest, we noted that sensilla with the greatest over-representation in females are ab10 (implied by greater numbers of Or49a/Or85f neurons) and ab3 (implied by greater numbers of Or22a/(Or22b) neurons) (Fig. 6B). Importantly, the latter example was previously overlooked due to underestimation of ab3 numbers quantified using an *Or22a*-Gal4 transgenic reporter (Grabe et al, 2016). The sexual

dimorphism in ab3 numbers is noteworthy because these neurons also display interspecific variation in number, notably representing the greatest difference of all Or neuron types between *D. melanogaster* and the ecological specialist *D. sechellia* (Auer et al, 2021), which has 2-3-fold more ab3 OSNs (Auer et al, 2020; Dekker et al, 2006; Takagi et al, 2024) (Fig. 6C). We recently provided evidence that increased OSN population size in *D. sechellia* enhances olfactory behavior not by increasing sensitivity of partner PNs, but rather by influencing their adaptation properties to repetitive or prolonged stimuli (Takagi et al, 2024). This invites the question of whether the dynamics of odor processing in PNs receiving input from ab3 and ab10 neurons are sexually dimorphic in *D. melanogaster* due to the differences in OSN number.

Shared sexually dimorphic and interspecific differences in OSN population size are not observed for other cell types. For example, while ab10 Or49a/Or85f neurons are over-represented in females, there is no species difference in ab10 (as inferred from Or67a OSN numbers) between *D. melanogaster* and *D. sechellia* (Fig. 6B,C). Reciprocally, while the ac3I Ir75b neuron population is greatly expanded in *D. sechellia* compared to *D. melanogaster* (Fig. 6C), it is of a similar size in males and females in both species (Prieto-Godino et al, 2017; Takagi et al, 2024).

*Relationship of glomerular size with neuron and synapse numbers:* previous studies suggested a shallow, but significant correlation between the number of OSNs and the size of the corresponding glomerulus (Grabe et al, 2016). We re-analyzed this relationship, both for all glomeruli where data are available, and those receiving input from Or and Ir OSNs separately (Fig. 6D). While we confirmed a statistically significant correlation overall, we found that this is driven by a strong relationship with Or glomeruli, as Ir OSN number and glomerular size are uncorrelated (Fig. 6D). These observations indicate that Ir glomerular size must be dictated by other properties.

Using the more extensive dataset from the FlyWire connectome (Dorkenwald et al, 2024; Schlegel et al, 2024), we therefore examined correlations between glomerular size and PN number, but there was no evidence of a strong relationship, globally or within either olfactory subsystem (Fig. 6E). However, comparison of glomerular size with the number of synapses that individual classes of OSNs make with PNs, LNs and other OSNs in the hemibrain connectome (Schlegel et al, 2021) revealed positive correlations in all cases, although this was only a trend for Ir glomeruli for OSN:PN synapses, potentially because of limited sample size (Fig. 6F–H). These observations indicate that the densities of OSN:PN, OSN:LN and OSN:OSN synapses are relatively consistent across glomeruli regardless of the number of input or output neurons. The determinant of Ir glomerular size differences remains an interesting open question, which might be answered by future analysis of other microarchitectural features revealed by the connectome.

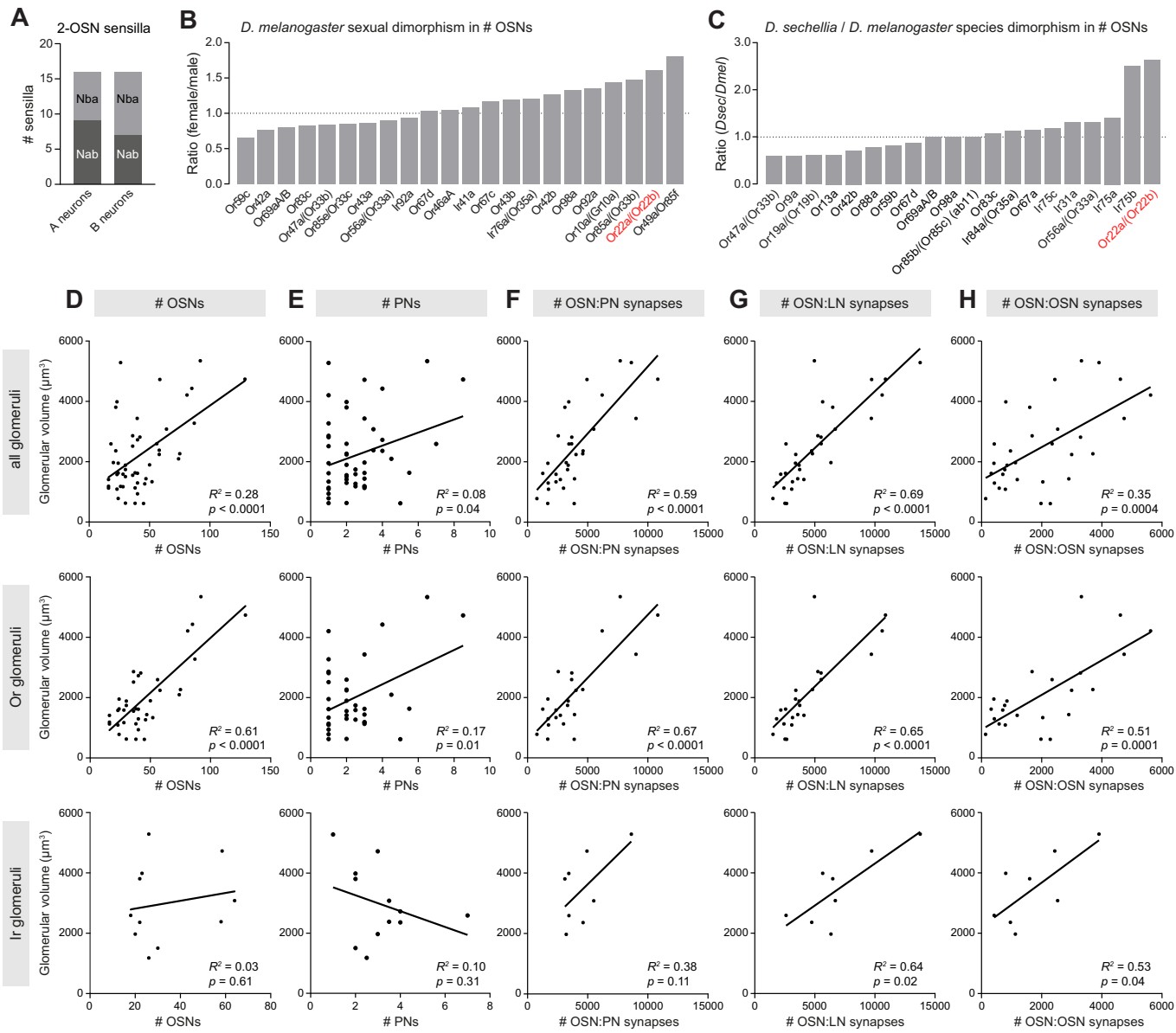

Figure 6. Organizational insights obtained from the integrated resource table.

(A) Stacked bar plot of the identity of OSN precursor type (Nab or Nba; Naa and Nbb are absent due to developmentally programmed cell death) in large-spike amplitude A and small-spike amplitude B neurons in sensilla with two OSNs. (B, C) Bar plots of the ratio of OSN numbers in female and male *D. melanogaster* (B) and female *D. sechellia* and *D. melanogaster* (C), revealing that the Or22a/(Or22b) population exhibits both sexual and species dimorphism. Note that only OSN populations for which direct experimental data are available (see Dataset EV1) are plotted; however, similar ratios can be inferred for the paired neurons within a given sensillum (e.g., Or85b/(Or85c) neurons in ab3 (Takagi et al, 2024)). For plots in (A–C), data are from Dataset EV1. (D) Correlation of glomerular volume and OSN numbers for all glomeruli (top), Or glomeruli (middle) and Ir glomeruli including the VC3 Or35a/Ir76a glomerulus (bottom). Note that OSN numbers per glomerulus were used; for nearly all populations, this number represents twice the number of OSNs per antenna because most OSNs project bilaterally. There are two exceptions (Ir75d and Gr21a/Gr63a OSNs), which project unilaterally; here the numbers of neurons per glomerulus are equivalent to those in the antenna. Simple linear regression used to determine coefficients of determination ($R^2$) and $P$ values: all glomeruli $R^2 = 0.28$, $P = 8e^{-5}$; Or glomeruli $R^2 = 0.61$, $P = 7e^{-9}$; Ir glomeruli $R^2 = 0.03$, $P = 0.6102$. (E) Correlation of glomerular volume and PN numbers for all glomeruli (top), Or glomeruli (middle), and Ir glomeruli (bottom). Simple linear regression: all glomeruli $R^2 = 0.08$, $P = 0.0405$; Or glomeruli $R^2 = 0.17$, $P = 0.0096$; Ir glomeruli $R^2 = 0.10$, $P = 0.3120$. (F) Correlation of glomerular volume and numbers of OSN:PN synapses for all glomeruli (top), Or glomeruli (middle) and Ir glomeruli (bottom). Simple linear regression: all glomeruli $R^2 = 0.59$, $P = 5e^{-7}$; Or glomeruli $R^2 = 0.67$, $P = 2e^{-6}$; Ir glomeruli $R^2 = 0.38$, $P = 0.1052$. (G) As in (F) for OSN:LN synapses. Simple linear regression: all glomeruli $R^2 = 0.69$, $P = 9e^{-9}$; Or glomeruli $R^2 = 0.65$, $P = 4e^{-6}$; Ir glomeruli $R^2 = 0.64$, $P = 0.0165$. (H) As in (F) for OSN:OSN synapses. Simple linear regression: all glomeruli $R^2 = 0.35$, $P = 0.0004$; Or glomeruli $R^2 = 0.51$, $P = 0.0001$; Ir glomeruli $R^2 = 0.53$, $P = 0.0418$. For all plots in (D–H), data are from Dataset EV1.

## Concluding remarks

Through identification of new olfactory sensory channels in *D. melanogaster*, we have "completed" our understanding of the basic molecular organization of this sensory system, notwithstanding structural and functional heterogeneity that undoubtedly exists within at least some sensory pathways. Using these findings as a stimulus to create an updated, integrated data resource of much of the enormous body of knowledge of the construction and function of this species' olfactory (as well as hygrosensory and thermosensory) systems, we believe this work should facilitate and inspire the coming years of research in the field.

# Methods

### Reagents and tools table

| Reagent/resource | Reference or source | Identifier or catalog number |
|---|---|---|
| **Experimental models** | | |
| *D. melanogaster* | | |
| *peb-Gal4* | Bloomington *Drosophila* Stock Center | RRID:BDSC_80570 |
| *w^1118* | Bloomington *Drosophila* Stock Center | RRID:BDSC_3605 |
| *Or13a-Gal4* | Bloomington *Drosophila* Stock Center | RRID:BDSC_23886 |
| *Or49b-Gal4* | Bloomington *Drosophila* Stock Center | RRID:BDSC_24614 |
| *UAS-mCD8::GFP* | Bloomington *Drosophila* Stock Center | RRID:BDSC_5130 |
| *Act5C-Gal4* | Bloomington *Drosophila* Stock Center | RRID:BDSC_4414 |
| *UAS-Ir76a^RNAi* (KK) | Vienna *Drosophila* Resource Center | #101590 |
| *UAS-Ir76a^RNAi* (TRiP) | Bloomington *Drosophila* Stock Center | RRID:BDSC_34678 |
| RNAi control (KK) | Vienna *Drosophila* Resource Center | #60100 |
| RNAi control (TRiP) | Bloomington *Drosophila* Stock Center | RRID:BDSC_36303 |
| **Oligonucleotides and other sequence-based reagents** | | |
| HCR RNA FISH probes | | |
| *Or7a* | Molecular Instruments | NM_078526.1 |
| *Or13a* | Molecular Instruments | NM_078635.3 |
| *Or19a* | Molecular Instruments | NM_080274.3 |
| *Or23a* | Molecular Instruments | NM_078734.4 |
| *Or35a* | Molecular Instruments | NM_165117.2 |
| *Or43a* | Molecular Instruments | NM_078923.3 |
| *Or46aA* | Molecular Instruments | NM_206072.2 |
| *Or46aB* | Molecular Instruments | NM_206071.2 |
| *Or47a* | Molecular Instruments | NM_078965.3 |
| *Or49b* | Molecular Instruments | NM_078997.3 |
| *Or56a* | Molecular Instruments | NM_079072.2 |

| Reagent/resource | Reference or source | Identifier or catalog number |
|---|---|---|
| *Or67b* | Molecular Instruments | NM_079283.5 |
| *Or69aA* | Molecular Instruments | NM_206348.1 |
| *Or69aB* | Molecular Instruments | NM_206347.1 |
| *Or82a* | Molecular Instruments | NM_164323.1 |
| *Or83c* | Molecular Instruments | NM_079520.3 |
| *Or98a* | Molecular Instruments | NM_079812.2 |
| *Ir76a* | Molecular Instruments | NM_001104177.3 |
| *Ir84a* | Molecular Instruments | NM_141463.2 |
| **Chemicals, enzymes, and other reagents** | | |
| Ammonia | Fisher Scientific | 7664-41-7 |
| Amylamine | Sigma-Aldrich | 110-58-7 |
| *E2*-hexenal | Sigma-Aldrich | 6728-26-3 |
| *2*-heptanone | Sigma-Aldrich | 110-43-0 |
| *1*-hexanol | Acros Organics | 111-27-3 |
| Hexyl acetate | Sigma-Aldrich | 142-92-7 |
| Indole | Sigma-Aldrich | 120-72-9 |
| *2*-methylphenol | Sigma-Aldrich | 95-48-7 |
| *3*-methylphenol | Sigma-Aldrich | 108-39-4 |
| *4*-methylphenol | Sigma-Aldrich | 106-44-5 |
| *1*-octen-3-ol | Acros Organics | 3391-86-4 |
| Paraffin oil (solvent) | Thermo Scientific | 8012-95-1 |
| Pentyl acetate | Sigma-Aldrich | 628-63-7 |
| Phenethylamine | Acros Organics | 64-04-0 |
| Phenylacetaldehyde | Alfa Aesar | 122-78-1 |
| **Software** | | |
| Prism v10.3.1 | http://www.graphpad.com | |
| RStudio | https://www.posit.co/download/rstudio-desktop | |
| 3D Slicer v5.6.2 | http://www.slicer.org | |
| LabChart Pro v8.1.5 | http://www.adinstruments.com | |
| Integrative Genomics Viewer (IGV) v2.18.4 | http://www.igv.org | |
| Seurat v4.3.0.1 | https://www.satijalab.org/seurat | |
| ggplot2 v3.4.3 | https://www.cran.r-project.org/web/packages/ggplot2/index.html | |

## RNA FISH

HCR RNA FISH was performed as described (preprint: Mermet et al, 2025) on a control *peb-Gal4* genotype (Figs. 1, 2 and EV1) or *w^1118* (Fig. 3), using 2–5-day-old female flies (see Reagent and Tool Table), cultured on standard wheat flour/yeast/fruit juice medium in incubators with 12 h light:12 h dark cycles at 25 °C. All probes were produced by Molecular Instruments (see Reagent and Tool Table). Images from antennae and maxillary palps were

acquired with confocal microscopes (Zeiss LSM710 or Zeiss LSM880 systems) using a ×40 (or ×63 for the palp) oil immersion objective and processed using Fiji software (Schindelin et al, 2012). For all histology experiments, the number of biological replicates quantified are shown in the figure legends; all expression patterns were confirmed in at least two independent technical replicates of these experiments.

## Electrophysiology

GFP-guided single-sensillum electrophysiological recordings were performed on 2–6-day-old females, cultured on a standard corneal/molasses/yeast medium in incubators with 12 h light:12 h dark cycles at 25 °C. Recordings used glass electrodes filled with sensillum recording solution, essentially as described (Vulpe et al, 2021). For ab6 sensilla we used *Or13a-Gal4/UAS-mCD8::GFP*; for ab11 sensilla we used *Or49b-Gal4/UAS-mCD8::GFP* (see Reagent and Tool Table). A Prior Scientific Lumen 200 Illuminator was used as the excitation light source. The sample was visualized using a BX51WI Olympus microscope with a ×1.6 magnification changer, a ×50 objective, and a Semrock GFP-4050B-OMF filter cube.

For *Ir76a* loss-of-function analysis in ac3 and ac4, we crossed the *Act5C-Gal4* driver to the following *Ir76a^{RNAi}* or RNAi control transgenic lines: *UAS-Ir76a^{RNAi}* (KK), *UAS-Ir76a^{RNAi}* (TRiP), RNAi control (KK), RNAi control (TRiP) (see Reagent and Tool Table). Female flies aged 3–7 days were used for recordings (see Source Data for Fig. 3 for final genotypes). ac3 and ac4 sensilla were identified based upon their stereotyped locations on the antenna (Benton et al, 2009) and their responses to diagnostic odors (Silbering et al, 2011).

Odorants (see Reagent and Tool Table) were diluted (v/v) in paraffin oil (or water for ammonia), as indicated in the figure plots. Odor cartridges were prepared by applying 50 μl odorant solution onto a Whatman 13-mm assay disc, which was inserted into a Pasteur pipette closed with a 1-ml pipette tip. Fly preps were placed in a 2 l/min air flow directed by a glass air tube. Odor stimuli were injected into the air flow for 0.5 s at 0.5 l/min. The odor response was calculated from the difference in OSN spike frequency (or summed frequencies of all OSNs for ac sensilla) in response to a 0.5 s odor puff compared to a 0.5 s solvent puff, as described (Vulpe et al, 2021). Experimenters were not blinded to genotype or stimulus when quantifying responses.

## Terminology

There is some inconsistency in the literature regarding the use of certain terms, which we aim to clarify here.

First, "Olfactory Receptor Neuron" (ORN) and "Olfactory Sensory Neuron" (OSN) terms have been used interchangeably. We have favored the latter, as the terminology "sensory" describes more generally the function of these neuron populations, rather than linking them to a molecular entity ("receptor"). Moreover, this general terminology better encompasses the diversity of sensory neuron types, which can express Ors, Irs, or Grs.

Second, the use of the terms "tuning receptor" and "co-receptor" are generally well-accepted, though not equally applicable in every neuron. "Tuning receptor" refers to the subunit defining stimulus-specificity of a sensory receptor complex, and likely directly binds and/or is conformationally modified by the stimulus. Some neurons

house multiple potential tuning receptors; the best-characterized case is the maxillary palp pb2 neuron expressing two functional receptors, Or85e and Or33c (Goldman et al, 2005). Several other examples of tuning receptor co-expression have been described, but when examined only one receptor was shown to be functional (e.g., the ab4 neuron expressing Or56a and Or33a, where only the former receptor appears to contribute to neuronal specificity (Stensmyr et al, 2012)). In this study, we indicate such potentially non-functional receptors in parentheses. "Co-receptors" are obligatory subunits necessary for olfactory receptor trafficking and function. Due to their broad expression across multiple classes of neurons, they are assumed not to contribute to the sensory specificity of a particular neuron type and likely do not bind ligands. While this is clearest for the Or co-receptor Orco, several Ir co-receptors exhibit narrower expression patterns in sets of neurons that respond to particular functional classes of stimuli (e.g., Ir76b in amine-sensing neurons; Ir93a in hygro/thermosensory neurons), and it cannot be excluded that such proteins have a more direct role in stimulus recognition. Many co-receptors are expressed in neurons where there is no corresponding tuning receptor (Task et al, 2022), but there is so far little evidence for their roles in such neurons (see also (preprint: Mermet et al, 2025)). Finally, tuning and co-receptor identity is ambiguous or irrelevant in certain neurons. For example, in aristal Gr28b.d neurons, this Gr appears to function alone (Mishra et al, 2018; Ni et al, 2013). The receptors in ab1C $CO_2$-sensing neurons, Gr21a and Gr63a, are each partially sufficient for conferring sensory responses, at least in *Xenopus* oocytes, although less effectively than when expressed together (Ziemba et al, 2023), and both are required for in vivo reconstitution of $CO_2$ sensitivity in heterologous neurons (Jones et al, 2007; Kwon et al, 2007).

Third, for sensillum nomenclature, we note the literature contains several discrepancies in the descriptions of the neuronal composition of ab6 and ai1 sensilla. The first characterization of ab6 was through electrophysiological recordings, which demonstrated the presence of two neurons: one responded to various alcohols (notably *1*-octen-3-ol) and the other to *4*-methylphenol (de Bruyne et al, 2001). Subsequent functional studies matched the response profile of Or49b receptors to ab6B neurons (Hallem et al, 2004). Further molecular and histological studies tentatively suggested Or49b is housed in the ab6 sensillum with Or85b and/or Or98b neurons (Couto et al, 2005). However, a later survey proposed that Or49b and Or13a neurons are paired in this sensillum, due to the close similarity of Or13a and ab6A response profiles (Galizia et al, 2010). This proposition was re-quoted in subsequent papers (e.g., (Auer et al, 2020; Grabe et al, 2016; Prieto-Godino et al, 2020)). Concurrently, targeted recording of sensilla housing Or13a neurons (through expression of GFP under the control of *Or13a-Gal4*) lead to its designation as the sole neuron housed in so-called ai1 sensilla, distinct from "ab6" sensilla housing Or49b neurons (Lin and Potter, 2015). However, the length of the putative ai1 sensillum resembles more closely small basiconic sensilla than other ai sensilla (Lin and Potter, 2015). Importantly, our recordings (Fig. 2B) unambiguously demonstrate the presence of a second neuron in this sensillum, which we have shown expresses Or46aB.

Recently, we demonstrated that Or49b-expressing neurons are paired with those expressing Or85b/(Or85c), and we described these as ab6 sensilla based on their expression of Or49b (Takagi et al, 2024). This receptor pairing might have been overlooked in previous studies because the majority of Or85b/(Or85c) neurons are housed in ab3, paired with Or22a/(Or22b) neurons (Takagi

et al, 2024). In the current study, we have determined that there are two sensilla populations that could potentially be named ab6: those housing Or49b and Or85b/(Or85c) neurons and those with Or13a and Or46aB neurons. We propose to give precedent to the original electrophysiological analysis (de Bruyne et al, 2001) by designating the ab6 sensillum as that housing Or13a and Or46aB neurons. The sensillum housing Or49b and Or85b/(Or85c) neurons therefore represents a new type of sensillum, which we name ab11. Finally, we note that one report described "ab11" and "ab12" sensilla, each housing three OSNs, one of which responds to the insect repellent citronellal (Kwon et al, 2010). The molecular identity of these sensilla is unclear, and they have not been described in any subsequent studies. Given the apparent completeness of the antennal lobe map with our discovery of Or46aB neurons, we suggest the sensilla classes described in that study represent variants of other basiconic classes (e.g., a three-OSN "abX" from (Nava Gonzales et al, 2021)), rather than new classes.

## Data resources and analysis

The snRNA-seq data and analysis methods are described in (preprint: Mermet et al, 2025); gene expression levels shown in the UMAPs are residuals from a regularized negative binomial regression, and have arbitrary units. The antennal lobe confocal images were re-analyzed and re-processed from a dataset generated in (Endo et al, 2007). The antennal lobe atlas used glomerular meshes previously generated by EM analysis of the antennal lobe (Bates et al, 2020), incorporating updated glomerular naming (Schlegel et al, 2021). Antennal lobe images and movies were generated using the open-source software 3D Slicer (Fedorov et al, 2012) (Fig. 5; Dataset EV2). Statistical analyses and plots were generated in RStudio with Seurat (v4.3.0.1) and GraphPad Prism 10.3.1. All other main sources of data are referenced directly in Dataset EV1.

## Data availability

Image and electrophysiological data are available in the Source Data files and the BioImage Archive: https://www.ebi.ac.uk/biostudies/BioImages/studies/S-BIAD1758.

The source data of this paper are collected in the following database record: biostudies:S-SCDT-10_1038-S44319-025-00476-8.

## Peer review information

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

## Acknowledgements

We are grateful to Roman Arguello for sharing bulk RNA datasets, to Chihiro Hama for sharing antennal lobe image data, and Veit Grabe and Silke Sachse for sharing anatomical data and discussions. We acknowledge the Bloomington *Drosophila* Stock Center (NIH P40OD018537) and the Vienna *Drosophila* Resource Center for fly stocks. We thank Tom Auer, Jamie Jeanne, Chris Potter, Lucia Prieto-Godino, Marcus Stensmyr, Chih-Ying Su, and members of the Benton and Menuz laboratories for feedback. Research was supported by NIH awards R35GM133209 and R21DC021267 to KM. Research in RB's laboratory is supported by the University of Lausanne, an ERC Advanced Grant (833548), and the Swiss National Science Foundation (310030 219185).

## Author contributions

**Richard Benton**: Conceptualization; Data curation; Formal analysis; Supervision; Funding acquisition; Validation; Investigation; Visualization; Writing—original draft; Project administration; Writing—review and editing; RB collated and analyzed most data for Dataset EV1. **Jérôme Mermet**: Formal analysis; Supervision; Investigation; Visualization; Methodology; Writing—review and editing; JM identified and characterized the Or46aA/B and Or35a/Ir76a cell types through snRNA-seq analysis and RNA FISH and contributed other OSN population quantifications. **Andre Jang**: Formal analysis; Investigation; Visualization; Writing—review and editing; AJ performed and analyzed electrophysiological experiments. **Keita Endo**: Investigation; Visualization; Writing—review and editing; KE provided data from SOP lineage labeling experiments. **Steeve Cruchet**: Investigation; SC performed and quantified RNA FISH experiments. **Karen Menuz**: Conceptualization; Data curation; Formal analysis; Supervision; Funding acquisition; Investigation; Visualization; Project administration; Writing—review and editing; KM contributed to data collation in Dataset EV1, analyzed bulk RNA-seq, and generated the antennal lobe atlas files.

Source data underlying figure panels in this paper may have individual authorship assigned. Where available, figure panel/source data authorship is listed in the following database record: biostudies:S-SCDT-10_1038-S44319-025-00476-8.

## Disclosure and competing interests statement

The authors declare no competing interests.

# Expanded View Figures

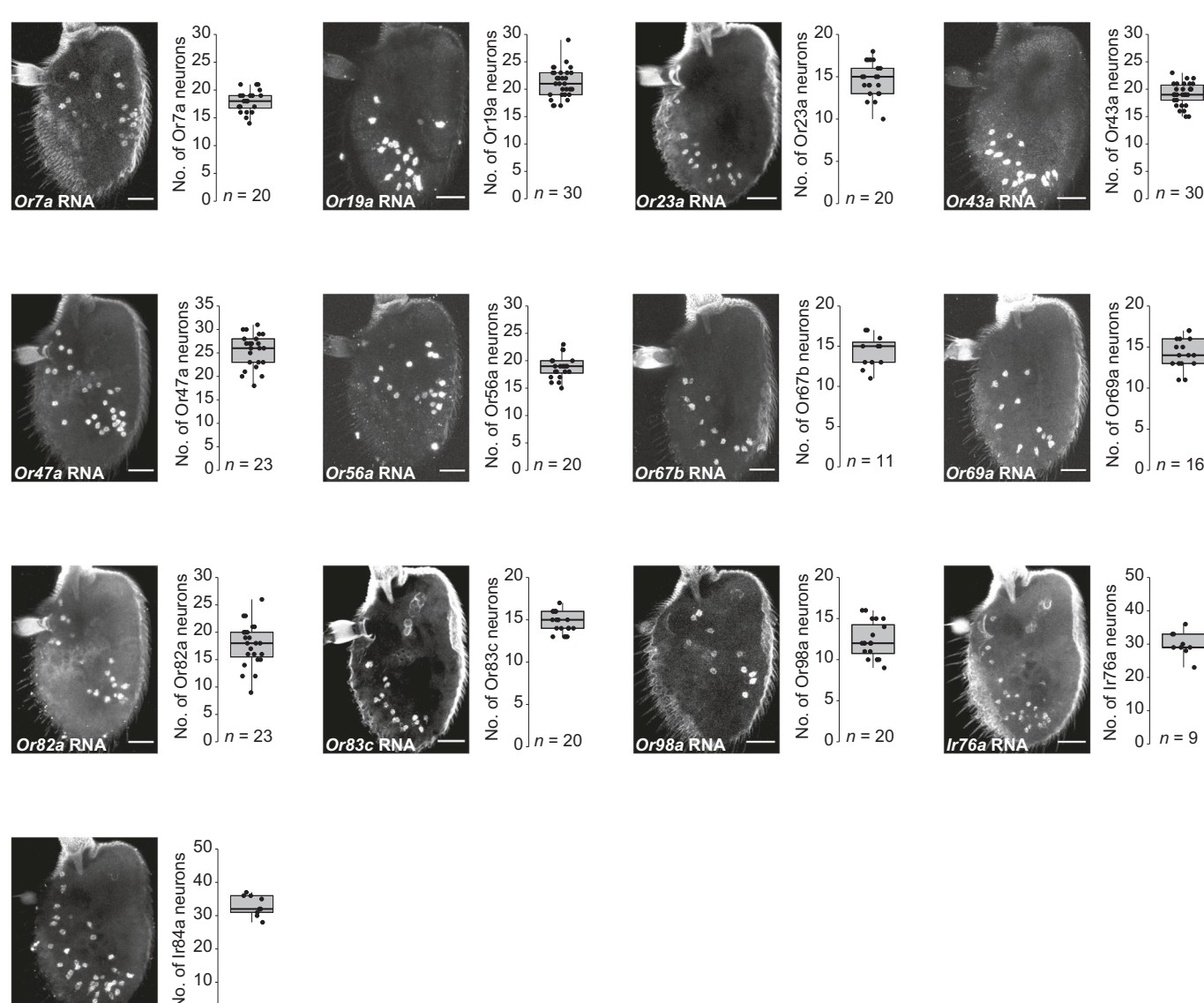

**Figure EV1.  Quantification of OSN populations.**

Representative images of HCR RNA FISH on whole-mount antennae (control genotype *peb-Gal4*) using the indicated gene probes, and quantifications of OSN population size. Box plots show median (thick line) and first and third quartiles, while whiskers indicate data distribution limits, overlaid with individual data points. These data are reported in Dataset EV1. For Or69aA/B neurons, the image shown is with an *Or69aA* probe, but the quantifications are pooled from images using either *Or69aA* or *Or69aB* probes. For Ir76a neurons, we only counted the cells with strong signal, which likely correspond only to those in ac4 sensilla. *n* is indicated underneath each box plot. Scale bars, 25 μm.

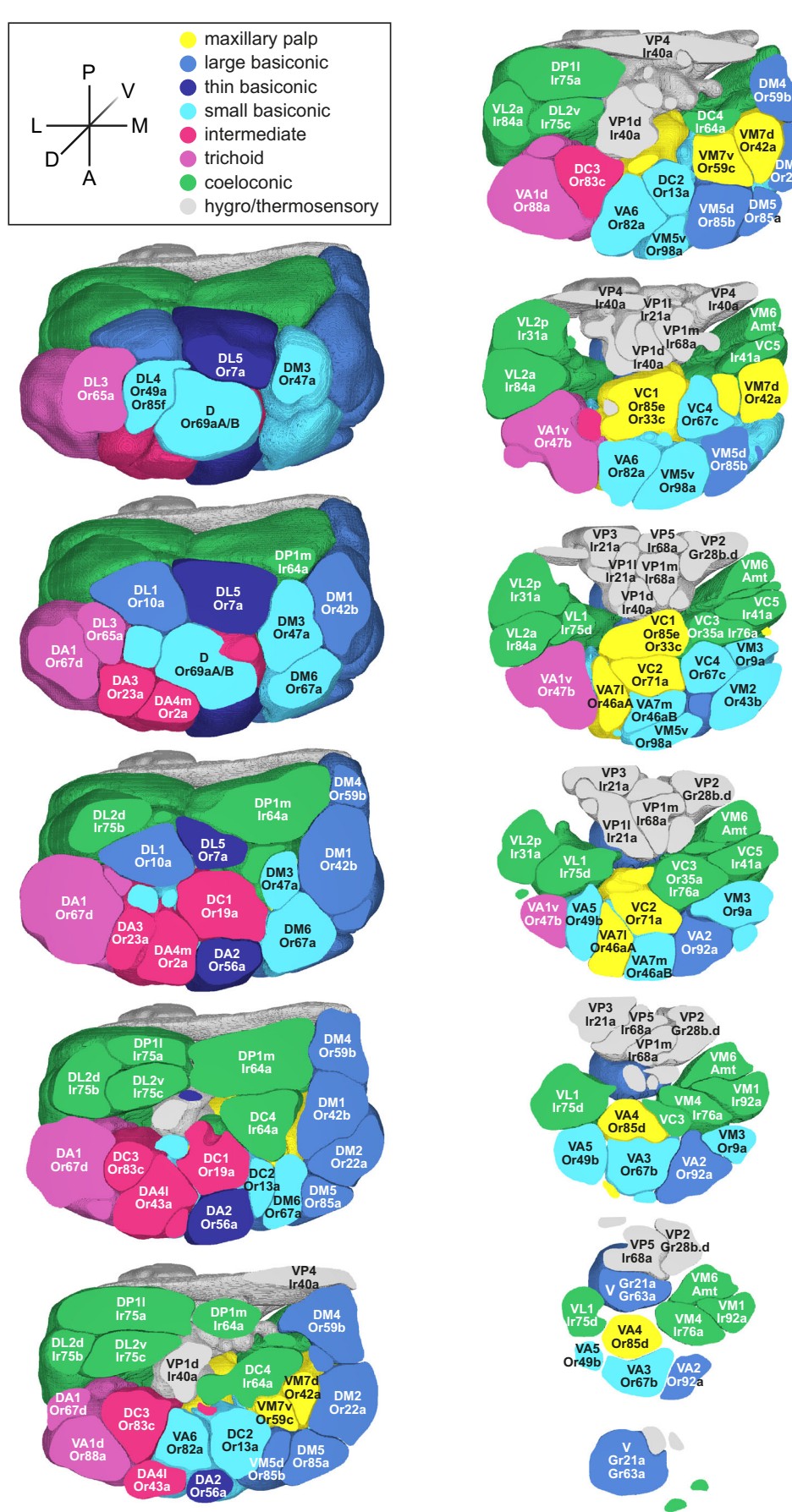

◄ **Figure EV2. Antennal lobe atlas of transverse sections.**

Transverse sections along the dorsal-ventral axis of an updated antennal lobe atlas, with coloring as in Fig. 5 and Dataset EV2. Such views are more typical of those obtained during in vivo calcium imaging experiments.

