## [Peer Review File · EMBO Reports]

An integrated anatomical, functional and evolutionary view of the *Drosophila* olfactory system

Richard Benton, Jérôme Mermet, Andre Jang, Keita Endo, Steeve Cruchet, and Karen Menuz

Corresponding author(s): Richard Benton (Richard.Benton@unil.ch) , Karen Menuz (karen.menuz@uconn.edu)

Review Timeline:

Submission Date:	16th Jan 25
Editorial Decision:	24th Feb 25
Revision Received:	1st Apr 25
Editorial Decision:	17th Apr 25
Revision Received:	22nd Apr 25
Accepted:	25th Apr 25

Transaction Report:

Dear Richard,

Thank you for the submission of your research manuscript to our journal. We have now received the full set of referee reports that is copied below.

As you will see, all three referees are very positive about your study and recommend publication at EMBO Reports after a few, rather minor, suggestions and concerns have been addressed.

Given these constructive and supportive comments, we would like to invite you to revise your manuscript with the understanding that the referee concerns (as detailed above and in their reports) must be fully addressed and their suggestions taken on board. Please address all referee concerns in a complete point-by-point response.

I agree with the suggestion from Referee 2 to clearly label the data from the Endo paper in Figure 2. You mention in the legend of Figure 2G that you re-processed the data from Endo et al., 2007. I assume that the images shown here are thus not the same as those shown in Endo et al but rather de novo generated using the data from the earlier study. Is this correct?

We usually recommend a revision within 3 months (May 24th). Please discuss the revision progress ahead of this time with the editor if you require more time to complete the revisions.

I am also happy to discuss the revision further via e-mail or a video call, if you wish.

You will find the general instructions on how to format your manuscript for EMBO Reports below, but let me first list a few things that are specific to your manuscript:

- Table S1 should be called "Dataset EV1". Maybe you could add a legend in a separate tab of the .xls file.
- "Data S1" and "Data S2" does not align with our naming conventions. Some of the data seem to be 'source data', i.e., raw data used to generate figures and graphs? If this is true, please include these data in the source data we ask for (see point 8 below). Otherwise, the files could be uploaded as Dataset EV#, Table EV#, Movie EV#.
- Data S2 could also be kept in the .zip format but then I suggest including a README.txt file with the legend and instructions.
- I am happy to discuss other formats and solutions that will be most useful for the field in reusing these data as a resource. If you choose an .xls format, please always include the legend in a separate tab of the file.
- For Figure S1 and S2 we have two options: either you include them as Figure EV# or you combine them to an Appendix. EV Figures are displayed in the html in a collapsible/expandable format. Their legends are part of the main manuscript, placed after the main figure legends. The Appendix is a single PDF that contains the figures and their legends. It needs a title page with a table of content and page numbers. The nomenclature is "Appendix Figure S#".
- Please supply a Reagents and Tools table (point 12 below).
- Please ensure that you have permission to publish Figure 1A, with the copyright at CSHL Press).

GENERAL INSTRUCTIONS:

- 1) a .docx formatted version of the manuscript text (including legends for main figures, EV figures and tables). Please make sure that the changes are highlighted to be clearly visible.
- 2) individual production quality figure files as .eps, .tif, .jpg (one file per figure).
Please download our Figure Preparation Guidelines (figure preparation pdf) from our Author Guidelines pages <https://www.embopress.org/page/journal/14693178/authorguide> for more info on how to prepare your figures.
- 3) a .docx formatted letter INCLUDING the reviewers' reports and your detailed point-by-point responses to their comments. As part of the EMBO Press transparent editorial process, the point-by-point response is part of the Review Process File (RPF), which will be published alongside your paper.

- 4) a complete author checklist, which you can download from our author guidelines (<https://www.embopress.org/page/journal/14693178/authorguide>). Please insert information in the checklist that is also reflected in the manuscript. The completed author checklist will also be part of the RPF.
- 5) Please note that all corresponding authors are required to supply an ORCID ID for their name upon submission of a revised manuscript (<https://orcid.org/>). Please find instructions on how to link your ORCID ID to your account in our manuscript tracking system in our Author guidelines (<https://www.embopress.org/page/journal/14693178/authorguide#authorshipguidelines>)
- 6) We replaced Supplementary Information with Expanded View (EV) Figures and Tables that are collapsible/expandable online. A maximum of 5 EV Figures can be typeset. EV Figures should be cited as 'Figure EV1, Figure EV2' etc... in the text and their respective legends should be included in the main text after the legends of regular figures.
- For the figures that you do NOT wish to display as Expanded View figures, they should be bundled together with their legends in a single PDF file called *Appendix*, which should start with a short Table of Content. Appendix figures should be referred to in the main text as: "Appendix Figure S1, Appendix Figure S2" etc. See detailed instructions regarding expanded view here: <https://www.embopress.org/page/journal/14693178/authorguide#expandedview>
 - Additional Tables/Datasets should be labeled and referred to as Table EV1, Dataset EV1, etc. Legends have to be provided in a separate tab in case of .xls files. Alternatively, the legend can be supplied as a separate text file (README) and zipped together with the Table/Dataset file.
- 7) Please include a dedicated "Data Availability" section at the end of the Methods (suggested wording: "The [structural coordinates | microarray | mass spectrometry] data from this publication have been deposited to the [name of the database] database [URL] and assigned the identifier [accession | permalink | hashtag]."). Should this not apply, this should still be stated as "This study includes no data deposited in external repositories."
- 8) At EMBO Press we ask authors to provide source data for the main figures. Our source data coordinator will contact you to discuss which figure panels we would need source data for and will also provide you with helpful tips on how to upload and organize the files.

Additional information on source data and instruction on how to label the files are available <https://www.embopress.org/page/journal/14693178/authorguide#sourcedata>.

10) Figure legends and data quantification:
The following points must be specified in each figure legend:

- the name of the statistical test used to generate error bars and P values,
- the number (n) of independent experiments (please specify technical or biological replicates) underlying each data point,
- the nature of the bars and error bars (s.d., s.e.m.)
- If the data are obtained from n {less than or equal to} 5, show the individual data points in addition to the SD or SEM.
- If the data are obtained from n {less than or equal to} 2, use scatter blots showing the individual data points.

See also the guidelines for figure legend preparation: <https://www.embopress.org/page/journal/14693178/authorguide#figureformat>

11) Our journal encourages inclusion of *data citations in the reference list* to directly cite datasets that were re-used and obtained from public databases. Data citations in the article text are distinct from normal bibliographical citations and should directly link to the database records from which the data can be accessed. In the main text, data citations are formatted as follows: "Data ref: Smith et al, 2001" or "Data ref: NCBI Sequence Read Archive PRJNA342805, 2017". In the Reference list, data citations must be labeled with "[DATASET]". A data reference must provide the database name, accession number/identifiers and a resolvable link to the landing page from which the data can be accessed at the end of the reference. Further instructions are available at <https://www.embopress.org/page/journal/14693178/authorguide#referencesformat>.

12) All Materials and Methods need to be described in the main text using our 'Structured Methods' format. According to this format, the Methods section includes a Reagents and Tools Table (listing key reagents, experimental models, software and relevant equipment and including their sources and relevant identifiers) followed by a Methods and Protocols section describing the methods, ideally using a step-by-step protocol format. The aim is to facilitate adoption of the methodologies across labs. Please download and fill our Reagents and Tools Table template (.docx), which you can find in our author guidelines: <https://www.embopress.org/page/journal/14693178/authorguide#structuredmethods>.

13) As part of the EMBO publication's Transparent Editorial Process, EMBO Reports publishes online a Review Process File to accompany accepted manuscripts. This File will be published in conjunction with your paper and will include the referee reports, your point-by-point response and all pertinent correspondence relating to the manuscript.

Kind regards,

Martina

=====

Referee #1:

Benton and colleagues present a comprehensive study of the *Drosophila melanogaster* olfactory system, integrating developmental, anatomical, and functional data into a unified and valuable resource. The authors identify a previously uncharacterized antennal sensory neuron population expressing Or46aB, completing the peripheral sensory map, and describe a novel "hybrid" olfactory sensory neuron class co-expressing functional odorant and ionotropic receptors. This work compiles and refines existing data on sensory neuron pathways, resolving inconsistencies in the literature and providing detailed visualizations of the antennal lobe and sensillar organization. Furthermore, the study examines the relationships between neuronal precursor identity, morphology, and glomerular size, offering new insights into the evolutionary and functional architecture of the fly olfactory system.

This manuscript, a well-written blend of review and original research with clear and informative figures, is outstanding. The authors have compiled an impressive dataset that will undoubtedly become a key resource for the field, and I am confident this will become the definitive reference for the *Drosophila* olfactory system.

In short, I have no objections; on the contrary, I enthusiastically support its publication!

Referee #2:

In this study, Benton and colleagues fill in remaining gaps in the very detailed description of the olfactory system of *Drosophila*, that combines developmental, anatomical, molecular, functional and behavioral data sets. Thus, in addition to the individual novel findings presented, this work strengthens the use of the fly olfactory system as a circuit of choice for studies of sensory processing. Thanks to the attached database, this manuscript will become a highly cited paper.

In terms of individual novel findings included, this study utilizes sn-RNA seq data presented elsewhere to show that the Or46a locus produces two transcripts Or46aA and Or46aB. By hybridization chain reaction RNA FISH the authors show that functional Or46aB is expressed in the neurons of the small basiconic sensilla ab6 in the antenna, while Or46aA is only expressed in the maxillary palp. Combining FISH and electrophysiological recordings, they provide convincing evidence to support the view that Or46aB and Or13a expressing neurons are paired in the ab6 sensilla, in contrast to previous assumptions. Second, they gather evidence indicating that Or49b and Or85b/(Or85c) receptors expressing neurons are housed in the ab11 sensilla. Third, considering previous data and the novel information that Or13a and Or46aB expressing neuron are housed in the same ab6 sensilla and arise from the same SOP- they also assign the VA7m antennal lobe glomerulus (the last orphan glomerulus) to the Or46aB expressing OSNs. Fourth, from RNAseq data included elsewhere, and functional recording of sensilla responses after RNAi, they indicate that one of the two neurons in the coeloconic sensillum ac3I co-expresses a Ir (Ir76a) and a Or (Or35a) receptor- the first report of Ir and Or co-expression in *Drosophila*.

All in all, this study closes the gap in our understanding of the olfactory circuit from olfactory sensory neurons to the antennal lobe in *Drosophila*. The database will be of great use to the community.

Few minor points should be addressed before publication:

1. To strengthen their data, the authors could add Or49b to their FISH experiments in Figure 2A, to show non-overlap with Or13a.
2. The data from the Endo paper in Figure 2 should be clearly marked as such - ideally also in the panels.
3. Figure 3C. The physiology data are convincing, but a rescue of the RNAi - or an alternative way to knock down or out Ir76a will be necessary to strengthen the genetic support for the authors' conclusion.
4. Figure 3A. Coexpression of Or35a and Ir76a should be highlighted.

Referee #3:

The authors completed the peripheral sensory map in the model organism *D. melanogaster*, generated an integrated dataset of these sensory neuron pathways, and used these dataset to reveal relationships between different organizational properties of this sensory system, and the new questions these stimulate. Here are my comments:

- Could the authors clarify in the text that ai2 and ai3 were previously named at2 and at3 or consider renaming them ai1 and ai2? If the naming remains ai2 and ai3, new students and trainees might assume the existence of an ai1. Nevertheless, what matters is the receptors these neurons express.
- For ac3I and ac3II, could the authors (if they agree) mention that, at the cellular level, it is difficult to distinguish them in wild-type *D. melanogaster*? Separating them requires genetic manipulation, as shown in Figure 3 of Prieto-Godino et al., 2017 (Neuron), or marking the specific sensillum type with GFP.
- Based on my experience, I have never recorded from an ab10 sensillum type where the B neuron is absent. Could the authors specify how often this occurs across individuals or whether this is a strain-specific issue? The same applies for at4B. This neuron is supposed to respond to CVA, but I have never observed this in my recordings.
- For the species with unique glomeruli, I'm not sure how useful it is to mention this, especially since there has been no validation or confirmation of these findings.

EMBOR-2025-61163V1: RESPONSE TO REVIEWERS

We thank the reviewers for their careful reading and constructive criticisms of our manuscript. Below, we provide responses to each of the raised issues.

Editorial requests

- I agree with the suggestion from Referee 2 to clearly label the data from the Endo paper in Figure 2. You mention in the legend of Figure 2G that you re-processed the data from Endo et al., 2007. I assume that the images shown here are thus not the same as those shown in Endo et al but rather de novo generated using the data from the earlier study. Is this correct?

RESPONSE: The images in Figure 2G-H are from a dataset that was presented only in the form of quantifications in Endo 2007 (doi:10.1038/n1832, Figure 3), along with quantifications from hundreds of other images. Thus, the images themselves were never previously published; indeed our new findings led us to re-analyse the earlier data enabling inference of the projection patterns of Or46aA neurons and previously-overlooked co-labelling of VA5 (Or49b) and VM5d (Or85b/(Or85c)) neurons, as described in the Results. In this scenario, we felt it was not necessary to explicitly label the figure panels themselves as we were not representing the actual raw data shown in the previous publication (and including a reference citation on a figure seems unusual). We reformulated the legend to "Data were re-analysed and re-processed from a dataset generated in (Endo et al., 2007)" to clarify our re-use of the data, but if you feel this remains ambiguous, we are happy to receive your advice on more explicit labelling.

- Table S1 should be called "Dataset EV1". Maybe you could add a legend in a separate tab of the .xls file.

RESPONSE: The file has been renamed. There is no legend for this table, mostly because we think it is fairly self-explanatory (and already includes many footnotes).

- "Data S1" and "Data S2" does not align with our naming conventions. Some of the data seem to be 'source data', i.e., raw data used to generate figures and graphs? If this is true, please include these data in the source data we ask for (see point 8 below). Otherwise, the files could be uploaded as Dataset EV#, Table EV#, Movie EV#.

RESPONSE: We have now renamed Data S2 as Dataset EV2, and re-assigned Data S1 to Source Data files.

- Data S2 could also be kept in the .zip format but then I suggest including a README.txt file with the legend and instructions.

RESPONSE: We have now included a README.txt file in this zipped data folder and removed the legend from the manuscript.

- I am happy to discuss other formats and solutions that will be most useful for the field in reusing these data as a resource. If you choose an .xls format, please always include the legend in a separate tab of the file.

RESPONSE: Please see comment above regarding the lack of a legend for the Dataset EV1.xls file.

- For Figure S1 and S2 we have two options: either you include them as Figure EV# or you combine them to an Appendix. EV Figures are displayed in the html in a collapsible/expandable format. Their legends are part of the main manuscript, placed after the main figure legends. The Appendix is a single PDF that contains the figures and their legends. It needs a title page with a table of content and page numbers. The nomenclature is "Appendix Figure S#".

RESPONSE: We have included these figures as Figure EV1 and EV2.

- Please supply a Reagents and Tools table (point 12 below).

RESPONSE: We have created a Reagents and Tools table, which includes some information originally in the Methods section.

- Please ensure that you have permission to publish Figure 1A, with the copyright at CSHL Press).

RESPONSE: We have now replaced this image with one from Dr Juergen Buerger, indicating the copyright as requested. I can supply an email granting the permission if necessary.

Referee #1

Benton and colleagues present a comprehensive study of the *Drosophila melanogaster* olfactory system, integrating developmental, anatomical, and functional data into a unified and valuable resource. The authors identify a previously uncharacterized antennal sensory neuron population expressing Or46aB, completing the peripheral sensory map, and describe a novel "hybrid" olfactory sensory neuron class co-expressing functional odorant and ionotropic receptors. This work compiles and refines existing data on sensory neuron pathways, resolving inconsistencies in the literature and providing detailed visualizations of the antennal lobe and sensillar organization. Furthermore, the study examines the relationships between neuronal precursor identity, morphology, and glomerular size, offering new insights into the evolutionary and functional architecture of the fly olfactory system.

This manuscript, a well-written blend of review and original research with clear and informative figures, is outstanding. The authors have compiled an impressive dataset that will undoubtedly become a key resource for the field, and I am confident this will become the definitive reference for the *Drosophila* olfactory system.

In short, I have no objections; on the contrary, I enthusiastically support its publication!

RESPONSE: We thank for the reviewer for their supportive comments.

Referee #2

In this study, Benton and colleagues fill in remaining gaps in the very detailed description of the olfactory system of *Drosophila*, that combines developmental, anatomical, molecular, functional and behavioral data sets. Thus, in addition to the individual novel findings presented, this work strengthens the use of the fly olfactory system as a circuit of choice for studies of sensory processing. Thanks to the attached database, this manuscript will become a highly cited paper.

In terms of individual novel findings included, this study utilizes sn-RNA seq data presented elsewhere to show that the Or46a locus produces two transcripts Or46aA and Or46aB. By hybridization chain reaction RNA FISH the authors show that functional Or46aB is expressed in the neurons of the small basiconic sensilla ab6 in the antenna, while Or46aA is only expressed in the maxillary palp. Combining FISH and electrophysiological recordings, they provide convincing evidence to support the view that Or46aB and Or13a expressing neurons are paired in the ab6 sensilla, in contrast to previous assumptions. Second, they gather evidence indicating that Or49b and Or85b/(Or85c) receptors expressing neurons are housed in the ab11 sensilla. Third, considering previous data and the novel information that Or13a and Or46aB expressing neuron are housed in the same ab6 sensilla and arise from the same SOP- they also assign the VA7m antennal lobe glomerulus (the last orphan glomerulus) to the or46aB expressing OSNs. Fourth, from RNAseq data included elsewhere, and functional recording of sensilla responses after RNAi, they indicate that one of the two neurons in the coeloconic sensillum ac3I co-expresses a Ir (Ir76a) and a Or (Or35a) receptor- the first report of Ir and Or co-expression in *Drosophila*. All in all, this study closes the gap in our understanding of the olfactory circuit from olfactory sensory neurons to the antennal lobe in *Drosophila*. The database will be of great use to the community.

Few minor points should be addressed before publication:

1. To strengthen their data, the authors could add Or49b to their FISH experiments in Figure 2A, to show non-overlap with Or13a.

RESPONSE: We have performed the requested double-FISH, revealing, as expected, no overlap (or pairing) between Or13a- and Or49b-expressing neurons. These data are shown in new Figure 2D.

2. The data from the Endo paper In Figure 2 should be clearly marked as such - ideally also in the panels.

RESPONSE: The images in Figure 2G-H are from a dataset that was presented only in the form of quantifications in Endo 2007 (doi:10.1038/n1832, Figure 3), along with quantifications from hundreds of other images. Thus, the images themselves were never previously published; indeed our new findings led us to re-analyse the earlier data enabling inference of the projection patterns of Or46aA neurons and previously-overlooked co-labelling of VA5 (Or49b) and VM5d (Or85b/(Or85c)) neurons, as described in the Results. In this scenario, we felt it was not necessary to explicitly label the figure panels themselves as we were not representing the actual raw data shown in the previous publication (and including a reference citation on a figure seems unusual). We reformulated the legend to “Data were re-analysed and re-processed from a dataset generated in (Endo et al., 2007)” to clarify our re-use of the data.

3. Figure 3C. The physiology data are convincing, but a rescue of the RNAi - or an alternative way to knock down or out *Ir76a* will be necessary strengthen the genetic support for the authors' conclusion.

RESPONSE: We would first like to highlight that we used two independent transgenic RNAi lines in our study; these cover different regions of the *Ir76a* gene, eliminating concerns of off-target effects. We showed quantifications for both of these (Figure 3D) but, for reasons of space, only provided representative traces for one of the lines (Figure 3C). Use of two lines is the gold standard for RNAi phenotypes in *Drosophila*. Moreover, these lines were validated in producing a very clear loss-of-function phenotype in ac4 sensilla (where *Ir76a* was previously shown to be expressed) and in ac3 sensilla (where we newly describe expression of *Ir76a* in the “hybrid” Or/Ir neuron). Lastly, the *Ir76a*^{RNAi} phenotypes are concordant with those of null mutants for the co-receptors genes *Ir76b* and *Ir25a* (as shown previously doi:10.3389/fncel.2021.759238).

The reviewer proposes two other suggestions: “rescue of the RNAi” would entail designing an RNAi resistant transgene, generating a new fly line, and several generations of breeding to obtain the experimental flies, which would take many months. A “knock out of *Ir76a*” would of course be the ideal reagent, but no sgRNA-encoding transgenes for this gene exist, so designing, creating and analysing such a mutant would also take minimum six months. (There is also a risk that deleting a portion of *Ir76a* would affect the function of a partially overlapping gene, CG14102, which encodes a ubiquitin ligase). Acknowledging this reviewer’s concern, we now include a statement regarding the future desirability to obtain an *Ir76a* mutant to strengthen the findings of our work.

4. Figure 3A. Coexpression of Or35a and Ir76a should be highlighted.

RESPONSE: We assume the reviewer is requesting a UMAP plot highlighting cells co-expressing Or35a and Ir76a, and provide such a plot in Reviewer Figure 1. Due to “drop-out” expression of *Or35a* and, particularly, *Ir76a*, in the ac3B neurons, only very few cells appear to co-express both receptor genes. However, such drop-out expression reflects the limited detection threshold of snRNA-seq (exacerbated here by the very low expression of *Ir76a*), rather than lack of functionally-relevant co-expression. This interpretation is supported by our FISH data in Figure 3B (illustrating the low *in situ* level of *Ir76a* transcripts) and functional analysis in Figure 3C-D showing that Ir-dependent amine responses are consistently observed in ac3B neurons (consistent with previous studies e.g. doi:10.1523/JNEUROSCI.2360-11.2011, doi:10.3389/fncel.2021.759238). We feel illustrating this co-expression plot in the figure only risks misleading readers unfamiliar with “drop out” expression to infer (incorrectly) that Or35a and Ir76a are mostly expressed in mutually-exclusive populations of ac3B neurons.

Reviewer Figure1. ac3B neurons co-express *Or35a* and *Ir76a*.

UMAPs of the ac3B neurons at different development phases extracted from the snRNA-seq atlas (Figure 1A), illustrating the individual expression and co-expression patterns of *Or35a* and *Ir76a* receptor genes. Cells expressing both *Or35a* and *Ir76a* are shown in yellow in the rightmost panel; this plot underestimates co-expression due to drop-out expression of both of the individual receptor genes.

Referee #3

The authors completed the peripheral sensory map in the model organism *D. melanogaster*, generated an integrated dataset of these sensory neuron pathways, and used these dataset to reveal relationships between different organizational properties of this sensory system, and the new questions these stimulate.

Here are my comments:

-- Could the authors clarify in the text that ai2 and ai3 were previously named at2 and at3 or consider renaming them ai1 and ai2? If the naming remains ai2 and

ai3, new students and trainees might assume the existence of an ai1. Nevertheless, what matters is the receptors these neurons express.

RESPONSE: We now include such a statement:

“...we distinguish the classes of antennal intermediate (ai2, ai3) and trichoid (at1, at4) sensilla more clearly, as these have been conflated in the past (e.g., in (Couto et al., 2005) ai2 and ai3 sensilla were referred to “at2” and “at3”, respectively).”

We prefer not to rename these sensilla “ai1” and “ai2” as the reviewer suggests because such name changes would likely lead to much confusion as some names would be ambiguous (e.g. “ai2” could refer to either Or83c/Or23a containing sensilla (ai2 in previous literature) or to those with Or19a/Or2a/Or43a (ai2 in future literature). Furthermore, the same sensillum type would then have up to three different names in the literature (e.g. “at3” (doi:10.1016/j.cub.2005.07.034) = “ai3” (current name) = ai2 (proposed name change by the reviewer). Additionally, by the same logic, “at4” should be renamed “at2”, which would only cause further confusion.

-- For ac3I and ac3II, could the authors (if they agree) mention that, at the cellular level, it is difficult to distinguish them in wild-type *D. melanogaster*? Separating them requires genetic manipulation, as shown in Figure 3 of Prieto-Godino et al., 2017 (Neuron), or marking the specific sensillum type with GFP.

RESPONSE: We agree and now make such a comment in the legend to Figure 3C, where we believe it is most pertinent.

-- Based on my experience, I have never recorded from an ab10 sensillum type where the B neuron is absent. Could the authors specify how often this occurs across individuals or whether this is a strain-specific issue? The same applies for at4B. This neuron is supposed to respond to CVA, but I have never observed this in my recordings.

RESPONSE: Indeed, the absence of the ab10B (Or85f) neuron – and the at4B (Or65a/b/c) neuron – in a subset of sensilla depends upon the strain (our unpublished data and doi:2025.01.16.632932 Figure 8J-K). We now indicate in the Figure 4 legend that the frequency of absence of these neurons is strain-dependent. The developmental basis and functional significance of such heterogeneity remain interesting open issues, beyond the scope of the current manuscript.

REDACTED: Reviewer Figure 2

-- For the species with unique glomeruli, I'm not sure how useful it is to mention this, especially since there has been no validation or confirmation of these findings.

RESPONSE: We have rephrased the footnote in the table (Dataset EV1) to “Glomeruli **putatively** unique to ...” to highlight this point. We agree there has not been independent validation of the findings reported in Depetris-Chauvin *et al.* 2023, and this is likely to be very hard as it would require identifying the corresponding receptors expressed in neurons innervating such glomeruli.

Dear Richard,

Thank you for the submission of your revised manuscript to EMBO Reports and for incorporating and addressing all of the minor concerns raised.

Before we can proceed with the official acceptance of your manuscript, I kindly ask you to address the following points.

- Please provide up to 5 keywords.
- Data availability section: please add the URL that resolves directly to the dataset deposited to BioImage Archive and please place the section before the Acknowledgments.
- Please update the 'Conflict of interest' paragraph to our new 'Disclosure and competing interests statement'. For more information see <https://www.embopress.org/page/journal/14693178/authorguide#conflictsofinterest>
- Regarding the Author Contributions, we now use CRediT to specify the contributions of each author in the journal submission system. Therefore, please remove the Author Contributions from the manuscript file and make sure that the author contributions in our online manuscript tracking system are correct and up-to-date. The information you specified in the system will be automatically retrieved and typeset into the article. You can enter additional information in the free text box provided, if you wish.
- References to preprints: please add (preprint:) before the in text citation and add [PREPRINT] at the end of the citation in the reference list. This applies to Tirian and Dickson 2017, Adavi et al 2024, and to Mermert et al 2025).
- When you re-analyse RNA-seq datasets from previous study, then please cite these as well. This would be in the format of our Data references. You first cite the study as a standard citation. In addition, you cite the dataset with Data ref: Author name et al... YEAR) in the text and in the reference list you add Author names.... YEAR, URL that resolves to the dataset [DATASET]. If possible, I would encourage you to add these citation. See also <https://www.embopress.org/page/journal/14693178/authorguide#referencesformat>.
- Our editorial policies do not allow to base conclusions on "data not shown". We noticed one instance on page 16. Please either remove the conclusion based on these data or add the data.
- Movies that are part of Dataset EV2: are these required to be part of this .zip folder? If so, please provide a legend for these and include them either as separate README.txt file or include it in the current one. If it is not absolutely necessary to have these movies included in the Dataset EV2 zip folder, they could also be uploaded separately as Movie EV#. In this case they would need their own README.txt file, which would then be zipped with the movie. In this case we would also need a callout to these in the text.
- Please remove the Supplementary Datasets section from the manuscript.
- Our production/data editors have asked you to clarify several points in the figure legends (see below). Please incorporate these changes in the manuscript and return the revised file with tracked changes with your final manuscript submission.
 - A) Statistical test information. Only p-values that are actually shown in the figure panel(s) should (and must) be defined in the legends, all others should be removed from (or added to) the legend. Moreover, we ask for the specification of exact p-values:
 - Please note that the exact p values are not provided in the legends of figures 1D, 3D, 6D, F, G.
 - Please indicate the statistical test used for data analysis in the legends of figures 6D-H.
 - B) Replicates and error bars:
 - Please note that the box plots need to be defined in terms of minima, maxima, centre, bounds of box and whiskers, and percentile in the legends of figure EV1
 - Please note that the error bars are not defined in the legend of figure 2A
- Finally, EMBO Reports papers are accompanied online by
 - A) a short (1-2 sentences) summary of the findings and their significance,
 - B) 2-3 bullet points highlighting key results and
 - C) a schematic summary figure that provides a sketch of the major findings (not a data image).Please provide the summary figure as a separate file in PNG or JPG format at a size of 550x300-600 pixels (width x height). Please note that the size is rather small and that text needs to be readable at the final size. Please send us this information along with the revised manuscript.

With kind regards,

Martina

The authors have addressed all minor editorial requests.

Prof. Richard Benton
University of Lausanne
Center for Integrative Genomics
Genopode Building
Lausanne 1015
Switzerland

Dear Richard,

Thank you very much for implementing the last minor edits. I am very pleased to accept your manuscript for publication in the next available issue of EMBO reports. Thank you for your contribution to our journal.

Kind regards,

Martina
